# Nav1.8-expressing neurons control daily oscillations of food intake, body weight and gut microbiota in mice
Clara Bullich-Vilarrubias[1,4], Marina Romaní-Pérez [1,4 ✉], Inmaculada López-Almela[1,3], Teresa Rubio[1], Carlos J. García[2], Francisco A. Tomás-Barberán [2] & Yolanda Sanz [1]

Recent evidence suggests a role of sensory neurons expressing the sodium channel Nav1.8 on the energy homeostasis control. Using a murine diphtheria toxin ablation strategy and *ad libitum* and time-restricted feeding regimens of control or high-fat high-sugar diets, here we further explore the function of these neurons on food intake and on the regulation of gastrointestinal elements transmitting immune and nutrient sensing.

The Nav1.8+ neuron ablation increases food intake in *ad libitum* and time-restricted feeding, and exacerbates daily body weight variations. Mice lacking Nav1.8+ neurons show impaired prandial regulation of gut hormone secretion and gut microbiota composition, and altered intestinal immunity. Our study demonstrates that Nav1.8+ neurons are required to control food intake and daily body weight changes, as well as to maintain physiological enteroendocrine and immune responses and the rhythmicity of the gut microbiota, which highlights the potential of Nav1.8+ neurons to restore energy balance in metabolic disorders.

The peripheral nervous system (PNS) allows the communication between the central nervous system (CNS) and the peripheral tissues. The afferent or sensory arm of PNS sends information from the periphery to the CNS while the efferent innervations transmit effector signals from the CNS to the peripheral tissues to maintain homeostasis.

Sensory ganglia contain the cell bodies of pseudo-bipolar sensory neurons. Spinal neurons of the dorsal root ganglia and vagal sensory neurons of the nodose ganglia transmit somatosensory and visceral information, respectively, to the CNS.

The majority of the spinal and vagal sensory neurons express the sodium channel Nav1.8[1,2]. Spinal sensory neurons expressing Nav1.8 are involved in nociception and pain[3,4], while Nav1.8+ vagal sensory neurons in the nodose ganglia mediate the viscerosensory transmission from lung and gastrointestinal tract as suggest prediction-based studies[5].

Targeted expression of dTomato in Nav1.8+ neurons in mice through Cre-LoxP-based methods reveals that Nav1.8-expressing afferents innervate the intestinal mucosa and the intraganglionic laminar endings (IGLEs) of the myentric plexus[1,6]. Complementary vagotomy experiments confirm the vagal origin of these afferents[1,7].

Moreover, Nav1.8+ nodose ganglion neurons express G protein coupled receptors for gut hormones, immunomodulatory lipids, microbial-derived metabolites and receptors involved in neurotransmitters signaling[8], suggesting a relevant interaction of these neurons with key gastrointestinal paths whereby energy and immune homeostasis are controlled. Indeed, spinal and vagal Nav1.8+ neurons are reported to protect against invasion, colonization and dissemination of enteric pathogens, by modulating intestinal mucosal immune cells[9]. All these evidences suggest that, beyond nociception and pain, intestinal spinal and/or vagal Nav1.8+ neurons might be important to mediate gastric distension, and nutrient and immune sensing from the intestine to the brain, suggesting their relevance in maintaining energy homeostasis in response to fat-enriched diets. In this regard, under an obesogenic diet, Nav1.8-expressing neurons are required to limit acute inflammation due to dietary lipids and have a modest role on meal patterns, since mice lacking these neurons do not show the reduced meal size and the increased feed rate detected in control littermates in response to high fat diet[10]. In addition, recent publications indicate that stimulation of IGLEs mechanoreceptors, highly expressing Nav1.8[1,6], reduces food intake after a meal by transmitting gastric distension to the

[1]Microbiome, Nutrition and Health Unit, Institute of Agrochemistry and Food Technology, National Research Council (IATA-CSIC), Valencia, Spain. [2]Quality, Safety and Bioactivity of Plant Foods, CEBAS-CSIC Murcia, Spain. [3]Present address: Research Group Intracellular Pathogens: Biology and Infection, Department of Animal Production and Health, Veterinary Public Health and Food Science and Technology, Faculty of Veterinary Medicine, Cardenal Herrera-CEU University, Valencia, Spain. [4]These authors contributed equally: Clara Bullich-Vilarrubias, Marina Romaní-Pérez. ✉e-mail: marina.romani@iata.csic.es

brain for further inhibiting AgRP-orexigenic neurons[6]. Preclinical studies that characterize genetic markers of sensory neurons combined with targeted neuronal manipulation identified specific functions of sensory neurons of the PNS[6,11,12]. This evidence would help to establish the basis for the design of new strategies to restore the disturbed neural control of the energy homeostasis in obesity and associated metabolic disorders. In this regard, a better understanding of how Nav1.8+ neurons maintain the energy homeostasis pre- and postprandially, particularly in response to high energy-dense foods and to interventions for weight loss, such as time-restricted feeding, might be key to identify novel strategies for restoring energy balance.

Here, we used Cre-LoxP technology to selectively ablate Nav1.8+ neurons in mice, allowing us to examine their role in energy balance. We show that Nav1.8+ neurons are required to control food intake and body weight fluctuations, as well as to maintain prandial regulation of gut hormone secretion, daily oscillations of gut microbiota and bile acid (BA) composition, and immunity under different dietary regimens, including *ad libitum* and time-restricted feeding under a control diet (CD) or a high-fat, high-sugar diet (HFHSD). Our results highlight the potential of Nav1.8+ neurons to restore energy homeostasis in metabolic disorders.

## Results
### Nav1.8-expressing neurons are required for weight gain and for food intake control after fasting in mice fed high-fat high-sugar diet

To evaluate the role of neurons expressing the ion channel Nav1.8 in key elements controlling energy homeostasis, we selectively depleted these neurons by a diphtheria toxin ablation strategy (as described in the Methods section) to obtain Nav1.8-deficient mice and control mice.

Since vagal afferents are involved in the energy homeostasis control, we examined whether the expression of Nav1.8 was cancelled in the nodose ganglion of the offspring. Compared with controls, mice expressing Cre,

hereinafter referred to as Nav1.8-cre/DTA mice or mice lacking Nav1.8+ neurons (Supplementary Figs 1a, b, 2a Cre⁻ or Cre⁺, for controls or Nav1.8-cre/DTA mice, respectively) showed a 28-fold decrease in *Scn10a* (coding for Nav1.8) mRNA levels in the nodose ganglion (Supplementary Fig. 2b), demonstrating the efficient genetic ablation of Nav1.8+ neurons in the vagal nerve.

The role of Nav1.8+ neurons in the control of body weight and food intake under an obesogenic diet was investigated (Supplementary Fig. 1a). Specifically, control and ablated mice fed on chow until 6 weeks of age were switched to HFHSD feeding during 8 weeks. After this period, Nav1.8-cre/DTA mice showed less body weight gain than their control littermates (Fig. 1a). The weight gain resistance to diet-induced obesity of mice lacking Nav1.8+ neurons was not due to lower food intake; in fact, contrary to what was expected from their lean phenotype, Nav1.8-cre/DTA tended to eat more than controls (Fig. 1b). Alternatively, we explored whether thermogenesis was involved in the genotype-related body weight differences. The quantification of cells expressing the thermogenesis marker uncoupling protein 1 (UCP1) in the brown adipose tissue (BAT) revealed that mice lacking Nav1.8-expressing neurons tended to have increased number of UCP1+ cells in BAT compared with controls ($p = 0.08$) (Fig. 1c).

Since the loss of Nav1.8-expressing neurons tended to dysregulate food intake and metabolic routes of fat utilization, we submitted mice to daily nutritional challenges to further explore the role of these neurons in the short-term control of food intake and body weight. Accordingly, we generated another cohort of mice (controls or Nav1.8-cre/DTA) that were initially fed *ad libitum* with either control diet (CD) or HFHSD, and then switched to 12 h of fasting followed by refeeding (12 h of restricted feeding) first in the light phase (LRF) for 3 weeks and then in the dark phase (DRF) for 3 weeks to explore the influence of feeding time or light/dark cues on feeding behavior and body weight variations (Supplementary Fig. 1b).

The analysis of body weight changes in response to 5 weeks of *ad libitum* feeding with CD or HFHSD revealed that Nav1.8-cre/DTA mice

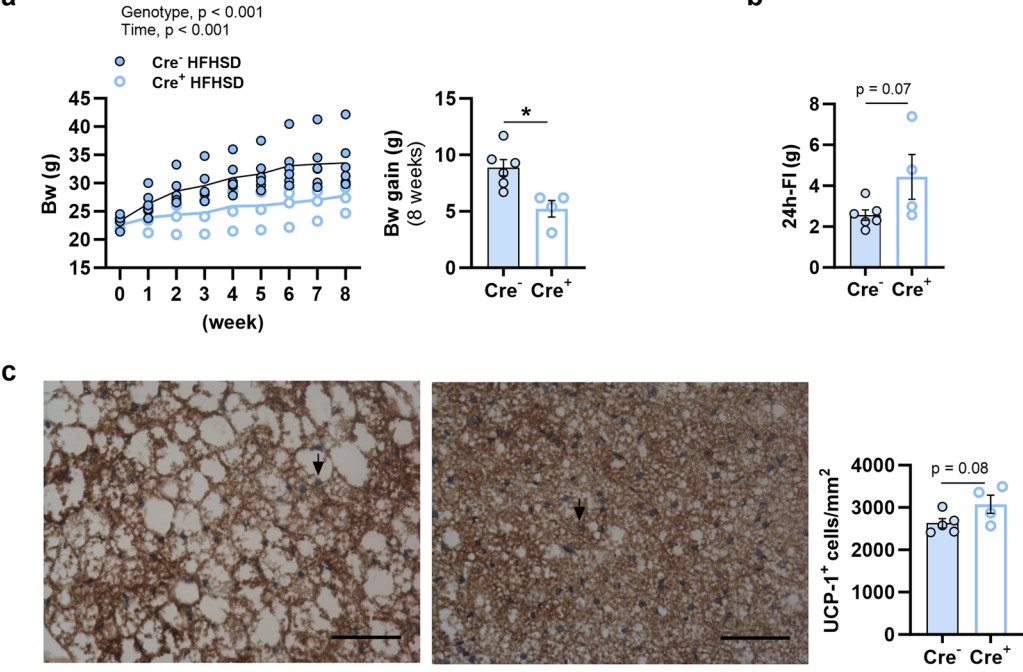

**Fig. 1 | Effects of the ablation of Nav1.8+ neurons on body weight, food intake and thermogenesis in mice fed high-fat high-sugar diet *ad libitum*.** In mice lacking Nav1.8+ neurons (Cre⁺) and their control littermates (Cre⁻) fed high-fat high-sugar diet (HFHSD) for 8 weeks, we measured: **a** Weekly body weight (Bw) and Bw gain; **b** 24 h- food intake (FI); and **c** Number of cells expressing the uncoupling coupled protein 1 (UCP-1) (graph) and immunostaining of UCP-1⁺ cells in brown adipose tissue (BAT), scale bar = 50 μm, black arrows indicate UCP1+ immunolabeling (in blue). Data are represented as scatter plots indicating individual values and the mean follow-up curve or scatter plots for individual values and bars with mean ± SEM (Cre⁻ HFHSD, $n = 6$-5 mice, and Cre⁺ HFHSD, $n = 4$ mice). Individual mice fed HFHSD are depicted by solid blue (Cre⁻) or empty blue (Cre⁺) circles; mean values in the follow-up curves in (**a**) are represented by black (Cre⁻) or bold blue (Cre⁺) lines. **a**: Two-way ANOVA with genotype (Cre⁻ or Cre⁺) and week as between-subject factors; **b**, **c**: unpaired Student´s t test. *$p < 0.05$.

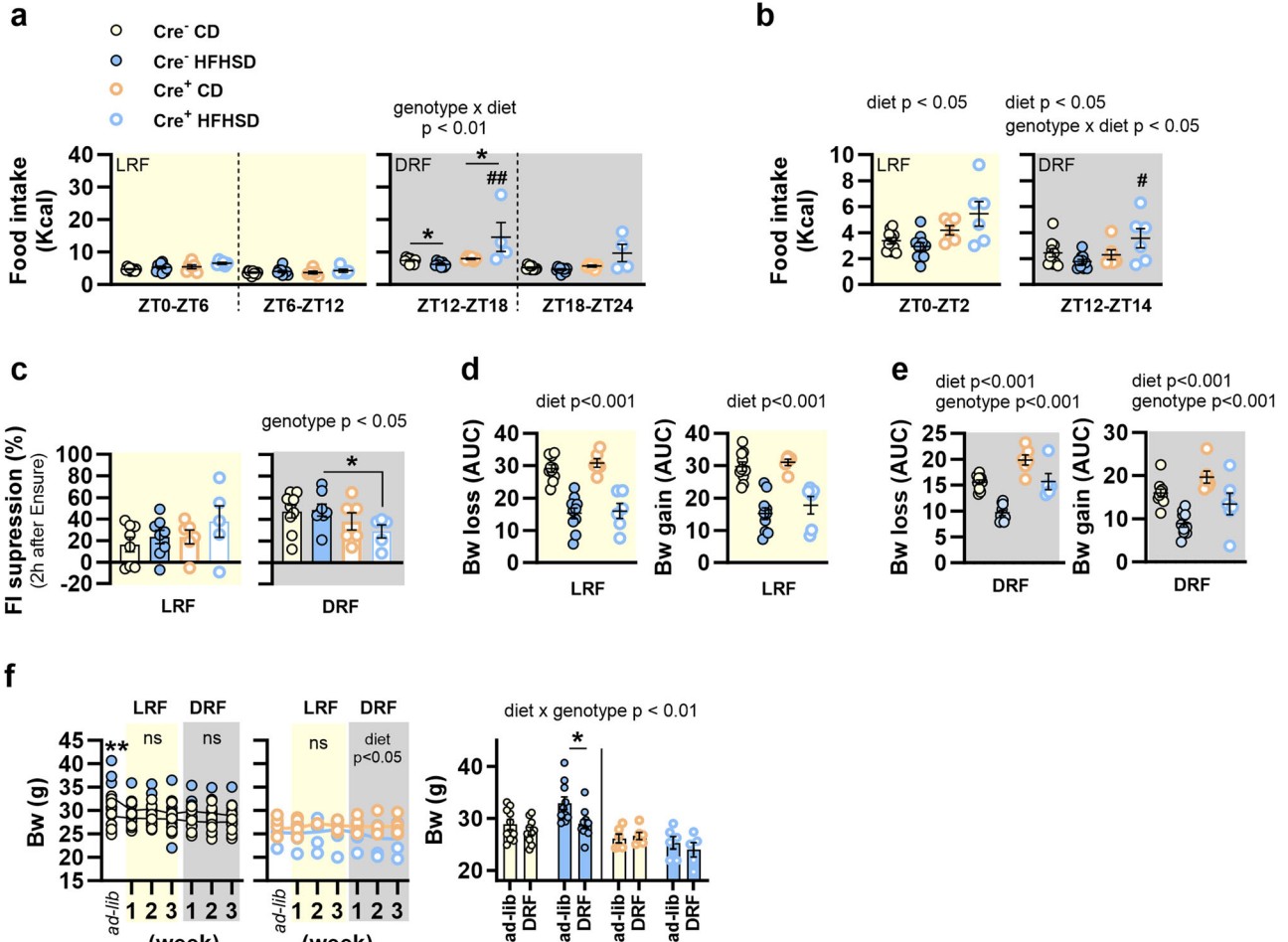

**Fig. 2 | Influence of the ablation of Nav1.8+ neurons in the control of food intake and body weight of mice under time-restricted feeding.** In mice lacking Nav1.8+ neurons (Cre+) and their control littermates (Cre−), fed either control or high-fat high-sugar diet (CD and HFHSD, respectively), we determined: **a** Food intake during the zeitgeber time (ZT)0 to ZT6 and ZT6-ZT12 in light restricted feeding (LRF) and during the ZT12-ZT18 and ZT18-ZT24 in dark restricted feeding (DRF). Data are the mean of 2 consecutive days at the end of each feeding restriction schedule. **b** Food intake during ZT0-ZT2 in the LRF and during ZT12-ZT24 in the DRF. Data are the mean of 3 consecutive days in the third week of the LRF or DRF. **c** Food intake (FI) suppression in the LRF or DRF represented as percentage of food intake after an oral gavage of Ensure® relative to FI after an oral gavage of saline (FI for either saline or Ensure® conditions was calculated using the mean of 3 consecutive days in the third week of the LRF or DRF). **d** Areas under the curve (AUC) of the Bw loss after 12 h of fasting and Bw gain after 12 h of refeeding during 3 weeks of LRF. **e** AUC of the Bw loss after 12 h of fasting and Bw gain after 12 h of refeeding during 3 weeks of DRF. **f** Bw follow-up in different feeding regimens: *ad libitum* (*ad-lib*), LRF and DRF; and impact of the time-restricted feeding on Bw (Bw is shown

after *ad-lib* and after DRF). Data are represented as scatter plots indicating individual values and bars with mean ± SEM or scatter plots for individual values and the mean follow-up curve (Cre− CD and Cre− HFHSD, *n* = 10 mice; Cre+ CD, *n* = 6 mice; Cre+ HFHSD, *n* = 5–7 mice). Mice fed CD are depicted by solid yellow (Cre−) or empty orange (Cre+) circles, mice fed HFHSD are depicted by solid blue (Cre−) or empty blue (Cre+) circles; mean values in the follow-up curves in (**f**) are represented by black (Cre− CD, Cre− HFHSD), bold orange (Cre+ CD) or bold blue (Cre+ HFHSD) lines; shadings in light yellow and grey represent LRF and DRF regimes, respectively; dashed lines separate the 6-h ZT timeframe. **a–f:** Two-way ANOVA with genotype (Cre− or Cre+) and diet (CD or HFHSD) as between-subject factors. Right graph of (**f**): Three-way ANOVA with genotype (Cre− or Cre+), diet (CD or HFHSD) and regimen (*ad libitum* or DRF) as between-subject factors. Main effects or interactions are indicated in the top of the graphs. When interactions were identified, differences between groups were assessed with Bonferroni's *post hoc* test. *$p < 0.05$, **$p < 0.01$ and ***$p < 0.001$ vs CD groups either Cre− or Cre+ (**a**, **f**), vs *ad libitum* feeding (**f**), or vs saline (**c**). #$p < 0.05$, ##$p < 0.01$ and ###$p < 0.001$ vs Cre− fed the same diet (HFHSD); ns non-significant.

showed reduced body weight gain especially under HFHSD. Indeed, an interaction between genotype and diet was identified on body weight gain (Supplementary Fig. 3a). *Post hoc* analysis indicated that Nav1.8-cre/DTA mice fed HFHSD, but not CD, had lower weight gain than equivalent controls (Supplementary Fig. 3a).

Daily food intake and body weight variations were assessed during restricted feeding. As feeding patterns of rodents show peaks at dawn and dusk[13], caloric intake was measured in two consecutive 6-h periods in the last 2 days of the LRF [zeitgeber time (ZT)0-ZT6 and ZT6-ZT12] and the DRF (ZT12-ZT18 and ZT18-ZT24). We observed that HFHSD-fed control mice ate less during the dusk peak (ZT12-ZT18) than equivalent CD-fed mice subjected to DRF, but not to LRF (Fig. 2a). In the dusk peak, we also identified an interaction between genotype and diet (Fig. 2a) showing that

Nav1.8-cre/DTA mice blunted the HFHSD-induced food intake reduction under DRF and ate more than controls (Fig. 2a). We replicated these results in the first 2 h of free access to food under LRF or DRF, when feeding behavior is governed by hunger signals after a period of fasting. In particular, a main diet effect indicated that HFHSD reduced food intake in control mice whatever the food-restricted regimen (LRF or DRF) (Fig. 2b), while the interaction between diet and genotype under DRF denoted that Nav1.8-cre/DTA mice fed HFHSD ate more than equivalent control mice (Fig. 2b).

We also explored food intake suppression in response to an oral load of Ensure® administered after 12 h of fasting by measuring 2 h of food intake at the onset of the refeeding period. Under DRF, but not in LRF, we identified a main effect of the genotype in food intake suppression (Fig. 2c). The analysis restricted to either CD or HFHSD indicated that the absence of Nav1.8+

especially inhibited the Ensure®-induced food intake suppression under HFHSD rather than under CD (Fig. 2c).

We also analyzed daily fasting-refeeding body weight loss and gain. Under LRF and compared with CD-fed mice, HFHSD-feeding induced resistance to weight loss or weight gain after 12 h of fasting or 12 h of refeeding, respectively, irrespective of the genotype (Fig. 2d and Supplementary Fig. 3b, c). Under DRF, in addition to the HFHSD effect on limiting the fasting-refeeding body weight variations, a genotype effect was also evident (Fig. 2e and Supplementary Fig. 3d, e), indicating that daily body weight loss and gain was higher in Nav1.8-cre/DTA mice than in control mice irrespective of the diet.

Analysis of the global effect of time-restricted feeding on body weight revealed that control mice fed HFHSD, but not fed CD, lost weight after 6 weeks of 12-h-restricted feeding (LRF followed by DRF) as compared with their weight under *ad libitum* feeding (Fig. 2f). By contrast, time-restricted feeding had no influence on the body weight of Nav1.8-cre/DTA mice fed either CD or HFHSD (Fig. 2f).

In summary, we demonstrated that the loss of Nav1.8+ neurons exacerbates food intake at dusk and the body weight loss after fasting, and attenuates the food intake suppression after a meal, especially under HFHSD feeding. Taking together, our findings suggest that these neurons mitigate hunger at the beginning of the active phase through mechanisms facilitating satiety and/or limiting the loss of energy stores during fasting. To note, the role of Nav1.8+ neurons on the short-term control of food intake seems to be governed by light/dark cues rather than feeding time since these effects were observed when food was provided in the dark phase, but not in the light phase.

## Nav1.8+ neurons modulate the intestinal immune system and affect gut microbiota by influencing its daily rhythms and pre- and postprandial oscillations

The gut microbiota shows food-entrainable diurnal oscillations[14] and, through its interaction with the diet, modulates feeding patterns by governing enteroendocrine signaling of the gut-brain axis[15–17]. In addition, innervations of the gastrointestinal tract establish a complex connection with the gut microbiota[18]. On these bases, we explored whether the ablation of Nav1.8+ neurons affects the gut microbiota composition with a particular emphasis in its pre- and postprandial oscillations during DRF, which in turn could affect feeding behavior.

The analysis of the influence of genotype and diet on the gut microbiota after fasting (ZT12) and refeeding (ZT18 and ZT24) showed that Nav1.8+ neurons were necessary to shape oscillations of alpha diversity, measured as Shannon index and InvSimpson index, as Nav1.8-cre/DTA mice fed either CD or HFHSD did not show significant prandial variations as controls (Fig. 3a, b). In CD-fed control mice, the Shannon and InvSimpson indices were reduced from fasting (ZT12) to 6 h of refeeding (ZT18) (Fig. 3a), whereas in HFHSD-fed controls, the Shannon index was lower in both postprandial periods (ZT18 and ZT24) than in the fasting period (ZT12) (Fig. 3b).

When exploring beta diversity under CD, the prandial condition rather than the genotype shaped two different microbiota communities in fasted mice and 6-h refed mice (Fig. 3c, circles for ZT12 and triangles for ZT18). In both genotypes, gut microbiota did not appear to shape a specific cluster after 12 h of refeeding (ZT24), but it was distributed across ZT12 and ZT18 clusters (Fig. 3c, smaller square dots). Nevertheless, HFHSD shaped different clusters depending on the genotype and the prandial condition. Indeed, in contrast to control mice, Nav1.8-cre/DTA mice did not exhibit different sample clustering in fasting and in 6 h of refeeding (Fig. 3d).

We also analyzed the main effects of genotype and zeitgeber time on amplicon sequence variants (ASVs) for each diet (CD and HFHSD), to identify bacterial taxa whose abundance varied across fasting and refeeding depending on the genotype. The genotype altered the abundance of some ASVs between refeeding periods (from ZT18 to ZT24) in CD-fed mice. The absence of Nav1.8+ neurons impeded the increase of the fecal abundance of *Bilophila* spp. and of ASVs from the families Ruminococcaceae,

Desulfovibrionaceae and Peptococcaceae from 6 h to 12 h of refeeding (Fig. 3e). Also, Nav1.8-cre/DTA mice showed postprandial variations in *Stenotrophomonas* spp., *Acinetobacter* spp., and *Dietzia* spp., with abundances higher in both postprandial periods than during fasting (ZT12 vs ZT18 or vs ZT24) (Fig. 3e). On HFHSD, Nav1.8+ ablation affected the prandial-related oscillations of an ASV from the family Desulfovibrionaceae and from *Alistipes* spp. Indeed, in contrast to control mice, the loss of Nav1.8+ neurons impeded the 12-h-postprandial increase of Desulfovibrionaceae compared with fasting (ZT12 vs ZT24). In addition, Nav1.8-cre/DTA mice did not show the decrease of *Alistipes* spp. after 6 h of refeeding (ZT12 vs ZT24) or an increase in its abundance after 12 h of refeeding (ZT18 vs ZT24) observed in controls (Fig. 3e).

We next questioned whether Nav1.8+ neurons impact the prandial variations of microbiota-derived metabolites. We focused on secondary bile acids (SBAs), which are biotransformed by the microbiota from primary bile acids (PBAs) synthesized by the host from dietary cholesterol[19] and secreted to the intestinal lumen to digest ingested food.

Under CD, the prandial-related variations in BAs were not significantly affected by genotype, except for an increase of cholic acid in fasted Nav1.8-cre/DTA mice compared with fasted mouse controls (Fig. 4a). By contrast, under HFHSD the fasting levels of SBAs were higher in control mice than in equivalent Nav1.8-cre/DTA mice, particularly deoxycholic, hyodeoxycholic, murideoxycholic, 3-oxo-chenodeoxycholic and ursodeoxycholic acids, followed by a progressive decrease in their levels in the postprandial period (Fig. 4b). The results indicate that, in controls but not in Nav1.8-cre/DTA mice, microbial BA metabolism was active, particularly in fasting conditions once PBAs generated in the prior refeeding period reach the distal gut.

We then assessed whether Nav1.8-expressing neurons are also involved in the daily fluctuations of the gut microbiota when mice have 24-h free access to food. Thus, after time-restricted feeding, mice were again switched to *ad libitum* regimen during 3 weeks.

The free access to HFHSD throughout the whole day again increased the body weight of control mice but not of Nav1.8-cre/DTA mice (Supplementary Fig. 4a, b). Importantly, under a HFHSD, the switch from restricted to *ad libitum* feeding reduced the survival of Nav1.8-cre/DTA mice (5 dead mice out of 7) and so, gut microbiota under *ad libitum* feeding conditions was only investigated in CD-fed mice.

We explored the role of Nav1.8+ neurons on the daily variations of alpha diversity in gut microbiota every 6 h, as well as on the rhythmicity of bacterial taxa. Alpha diversity, measured as observed ASVs, Shannon index and InvSimpson index, was affected by the zeitgeber time but not by the genotype, with a reduction from the light phase to the beginning of the dark phase, followed by an increase at the end of this phase (Fig. 5a). Using the JTK_cycle algorithm, we found that Nav1.8+ neurons were critical to maintain the daily rhythmicity of some ASVs (Fig. 5b). Specifically, in controls, but not in Nav1.8-cre/DTA mice, we detected five rhythmic ASVs that were higher in abundance in the dark phase, including those belonging to the order Gastranaerophilales or species from the genera *Alistipes* or *Colidextribacter* (Fig. 5c). ASVs from the families Muribaculaceae and Oscillospiraceae also showed daily rhythms in control mice (Fig. 5c).

Since gut microbiota and intestinal immunity are closely related[20], we then examined whether these microbiota changes were associated with alterations in the gene expression of gut barrier markers and in the abundance of immune cells in the intestinal mucosa. The gene expression of several markers of intestinal barrier defence and integrity were lower in Nav1.8-cre/DTA mice than in control mice, including the ileal levels of *Tcf4*, involved in Paneth cell development, and the colonic levels of the tight junction proteins *Ocln* and *Cldn3* (Fig. 5d). Gene expression of antimicrobial peptides (*DefA*, *Lyz1*, *Reg3g*) or proliferation (*Ki67*) markers in ileum and colon was unaffected by the genotype (Supplementary Fig. 5a). The abundance of natural killer cells (NK) and type 1 innate lymphoid cells (ILC1) in the small intestinal epithelium was unaffected by the genotype (Supplementary Fig. 5b). Contrastingly, analysis of the lamina propria revealed a reduction in the abundance of type 2 ILC (ILC2), involved in type

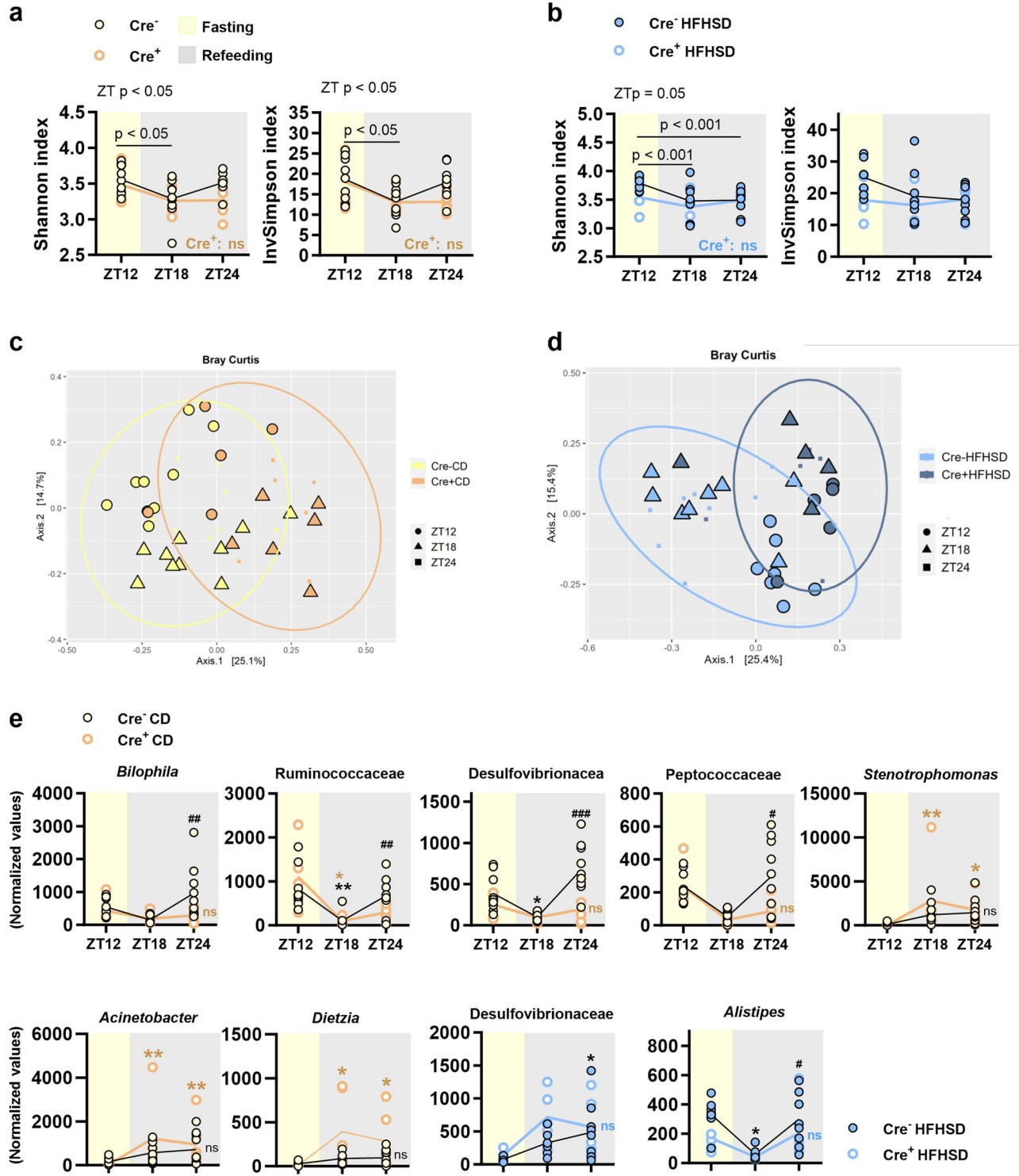

2-like responses and tissue repair (Fig. 5e), and regulatory T cells (Treg) (Fig. 5f), critical to control inflammation and to develop immune tolerance, in Nav1.8-cre/DTA mice, and an increase in T helper 17 cells (Th17) (Fig. 5g), suggesting disrupted intestinal immune homeostasis. Other immune cells such as type 3 ILC (ILC3) or type 1 and type 2 macrophages (M1 and M2) were unaffected by the absence of Nav1.8+ neurons (Supplementary Fig. 5b). Notably, spleen weight was greater in Nav1.8-cre/DTA mice than in control mice (Supplementary Fig. 5c), suggesting systemic inflammation, although no changes were observed in the splenic abundance

of memory or effector T cells or macrophages between genotypes, except for a trend for a decrease in M2 in Nav1.8-cre/DTA mice (Supplementary Fig. 5d).

We next addressed how the loss of Nav1.8+ neurons could affect intestinal inflammation. Thus, on the basis that Nav1.8+ neurons express VIP[21], a neural-produced peptide with immunomodulatory properties, we next examined the duodenal and ileal levels of this neuropeptide. VIP protein was undetectable in the duodenum. In the ileum, we found a trend for decreased VIP levels in Nav1.8-cre/DTA mice; indeed, while most of the

**Fig. 3 | Effects of the ablation of Nav1.8+ neurons on gut microbiota composition during dark-restricted feeding.** At the end of the dark restricted feeding, in mice lacking Nav1.8+ neurons (Cre+) and their control littermates (Cre-), fed either control or high-fat high-sugar diet (CD and HFHSD, respectively), we determined: **a**, **b** Alpha diversity after 12 h of fasting (ZT12) and 6 and 12 h of refeeding (ZT18 and ZT24) measured with the Shannon and Inverse Simpson indices. **c**, **d** Beta diversity after 12 h of fasting (ZT12) and 6 and 12 h of refeeding (ZT18 and ZT24), determined by principal coordinate analysis (PCoA) (phyloseq::ordinate function and "bray" distance). **e** Normalized data using trimmed mean of M values (TMM) of the abundance of amplicon sequence variants (ASVs) after 12 h of fasting (ZT12) and 6 and 12 h of refeeding (ZT18 and ZT24). Data are represented as scatter plots indicating individual values and the mean follow-up curve (Cre- CD and Cre- HFHSD, *n* = 9 mice; Cre+ CD, *n* = 5–6 mice; Cre+ HFHSD, *n* = 5 mice). Mice fed CD are depicted by solid yellow (Cre-) or empty orange (Cre+) circles, mice fed HFHSD are depicted by solid blue (Cre-) or empty blue (Cre+) circles; mean values in the follow-up curves in (**a**, **b**, **e**) are represented by black (Cre- CD, Cre- HFHSD), bold orange (Cre+ CD) or bold blue (Cre+ HFHSD) lines; shadings in light yellow and grey represent fasting (light phase) and refeeding (dark phase) periods, respectively; in (**c**, **d**), circles, triangles and squares represent ZT12, ZT18 and ZT24, respectively, mice on CD feeding are represented in solid yellow (Cre-) or solid orange (Cre+), mice on HFHSD feeding are represented in solid light blue (Cre-) or solid dark blue (Cre+). **a**, **b**: Two-way ANOVA with genotype (Cre- or Cre+) and zeitgeber time (ZT) as between-subject factors followed by Bonferroni's *post hoc* test. $p < 0.05$, $p < 0.01$ and $p < 0.001$ indicate significant differences between ZT within the same experimental group; **e**: quasi-likelihood F-tests and Bonferroni *post hoc* correction. #$p < 0.05$, ##$p < 0.01$ and ###$p < 0.001$ vs ZT18. *$p < 0.05$ and **$p < 0.01$ vs ZT12 (black for Cre- and orange for Cre+). ns: non-significant differences.

control mice showed detectable levels of VIP most Nav1.8-cre/DTA mice had undetectable levels (Supplementary Fig. 5e). We reasoned that the absence of Nav1.8+ neurons might impact brain-to-gut communication through the sympathetic and parasympathetic efferent tone, influencing the intestinal inflammatory profile. The sympathetic tone in the duodenum, but not in the distal regions, was enhanced in Nav1.8-cre/DTA mice, as indicated by the higher levels of noradrenaline (NE) (Fig. 5h). By contrast, the levels of acetylcholine (Ach) were unaffected (Fig. 5h).

Altogether, these analyses establish that Nav1.8+ neurons are relevant to shape daily oscillations of the gut microbiota. Especially under DRF of HFHSD, Nav1.8+ neurons govern the postprandial decrease of the alpha diversity and SBAs production at the beginning of the active phase. The function of Nav1.8+ neurons regulating gut microbiota could be attributed to their effects as modulators of immune cells in the intestine, which could be in turn modulated by a balanced efferent tone.

### Nav1.8+ neurons are required for the secretion of gut hormones involved in hunger and satiety signaling

Next, we assessed whether the altered daily rhythms of the gut microbiota of Nav1.8-cre/DTA mice were aligned with altered secretion of gut hormones as primary drivers of short-term control of food intake.

At the end of the experiment, we measured gut hormones during fasting and after an oral load of Ensure® (Fig. 6a–f) in mice fed CD *ad libitum*, since, as mentioned above, most Nav1.8-cre/DTA mice did not survive a final period of *ad libitum* HFHSD feeding.

We identified interactions between genotype and prandial condition (fasting and post-Ensure® exposure) on gut hormones, predominantly those secreted in the upper small intestine such as ghrelin and GIP (Fig. 6a and b). In particular, the fasting levels of ghrelin were lower in Nav1.8-cre/DTA mice than in control mice, with no effect in response to Ensure® (Fig. 6a). Contrastingly, the Ensure®-induced secretion of GIP was boosted in Nav1.8-cre/DTA mice, whereas fasted levels remained unaffected by the genotype (Fig. 6b). CCK levels remained unchanged irrespective of the genotype or the prandial condition (Fig. 6c). The distal gut anorexigenic hormones PYY and GLP-1 were also affected by the genotype (Fig. 6d, e). Whereas PYY levels were higher in Nav1.8-cre/DTA mice whatever the prandial condition (Fig. 6d), the genotype and the prandial condition showed a trend for interaction on GLP-1 (Fig. 6e). Analysis of GLP-1 restricted to either fasting or oral Ensure® revealed that fasting GLP-1 levels were higher in Nav1.8-cre/DTA mice than in controls (Fig. 6e). Comparable results were found for neuropeptide αMSH (Fig. 6f), with a significant interaction between genotype and prandial condition (Fig. 6f). Analysis of the slope between the fasting and Ensure® states for GLP-1 and αMSH levels (Fig. 6g) supported the defective Ensure®-induced increase of these peptides in Nav1.8-cre/DTA mice (Fig. 6e, f), as the slopes of both peptides were lower than those of controls (Fig. 6g).

The ileal and colonic transcript levels of PYY and GLP-1 (*Pyy* and *Proglucagon*) were unchanged between genotype (Supplementary Fig. 6a); however, the expression of the transcriptional activator *Neurod1* was lower in ileum and *Ngn3* mRNA levels remained unchanged in duodenum, ileum and colon (Fig. 6h), suggesting downregulated *Neurod1*-mediated ileal L-cell differentiation.

In mice lacking Nav1.8+ neurons, we also found disturbances in the gene expression of hypothalamic markers involved in food intake control. Specifically, the gene expression of the orexigenic neuropeptide *Npy* was higher in Nav1.8-cre/DTA mice than in controls, whereas the transcripts of receptors involved in transmitting anorexigenic signals such as CCK, leptin, GLP-1 and αMSH receptors (*Cck1r*, *Lepr*, *Glp1r* and *Mc4r*) were lower (Fig. 6i). Other hypothalamic neuropeptides such as *Agrp*, *Pomc* or *Cart* remained unaffected (Supplementary Fig. 6a).

Food intake and body weight variations throughout the day were also analysed in these mice. Analyses restricted to each phase indicated that, in the dark phase (ZT12-ZT24), Nav1.8-cre/DTA mice had higher caloric intake than their control littermates, with no change in the light phase (ZT0-ZT12) (Fig. 7a), although 24 h-food intake (ZT0-ZT24) was not affected by the ablation of Nav1.8+ neurons (Fig. 7a). We also measured food intake and body weight every 6 h during 24 h in line with the bimodal pattern of nocturnal food intake in rodents. Food intake was higher in Nav1.8-cre/DTA mice than in controls at dusk (ZT12-ZT18) (Fig. 7b), and this was accompanied by a higher body weight loss in the first 6 h of the light period (ZT0-ZT6) (Fig. 7c). In addition, the positive correlation between 24-h body weight variation and 24-h food intake in control mice was lost in Nav1.8-cre/DTA mice (Fig. 7d), suggesting that Nav1.8+ neurons are required to properly control body weight in response to food intake and vice versa. Supporting this, the weight of epididymal white adipose tissue (eWAT), but not that of inguinal WAT (iWAT) or BAT, was lower in Nav1.8-cre/DTA mice than in control mice (Fig. 7e), as were the plasma levels of leptin, irrespective of the prandial state (fasting or after an oral load of the liquid mixed nutrient solution, [Ensure®]) (Fig. 7f). Gene expression analysis of eWAT revealed a trend for higher expression of the beta 3 adrenergic receptor, *Adrb3*, which stimulates the noradrenergic-mediated fat mobilisation, in Nav1.8-cre/DTA mice than in control mice, whereas *Adrb2* expression was unchanged (Fig. 7g). In line with the apparent energy deficient status, Nav1.8-cre/DTA mice had reduced fasting glycemia (Fig. 7h) independently of changes in plasma insulin and glucagon levels, which were unaffected in Nav1.8-cre/DTA mice (Supplementary Fig. 6b, c).

Taken together, these findings indicate that even when mice are fed CD *ad libitum*, Nav1.8-expressing neurons are required for controlling food intake at the beginning of the active phase and for body weight loss in the resting phase. Nav1.8+ neurons could govern feeding patterns by influencing gut-to-brain transmission of gut hormone-mediated orexigenic and anorexigenic signaling.

## Discussion

Using *ad libitum* feeding and food restriction regimens, we identify a role for sensory neurons expressing the Nav1.8 sodium channel in the control of body weight and food intake. We also show that these neurons are relevant for maintaining the gut-brain endocrine pathways transmitting hunger and satiety signals, and in shaping the pre- and postprandial oscillations of the gut microbiota. At the long-term, these neurons are required to show

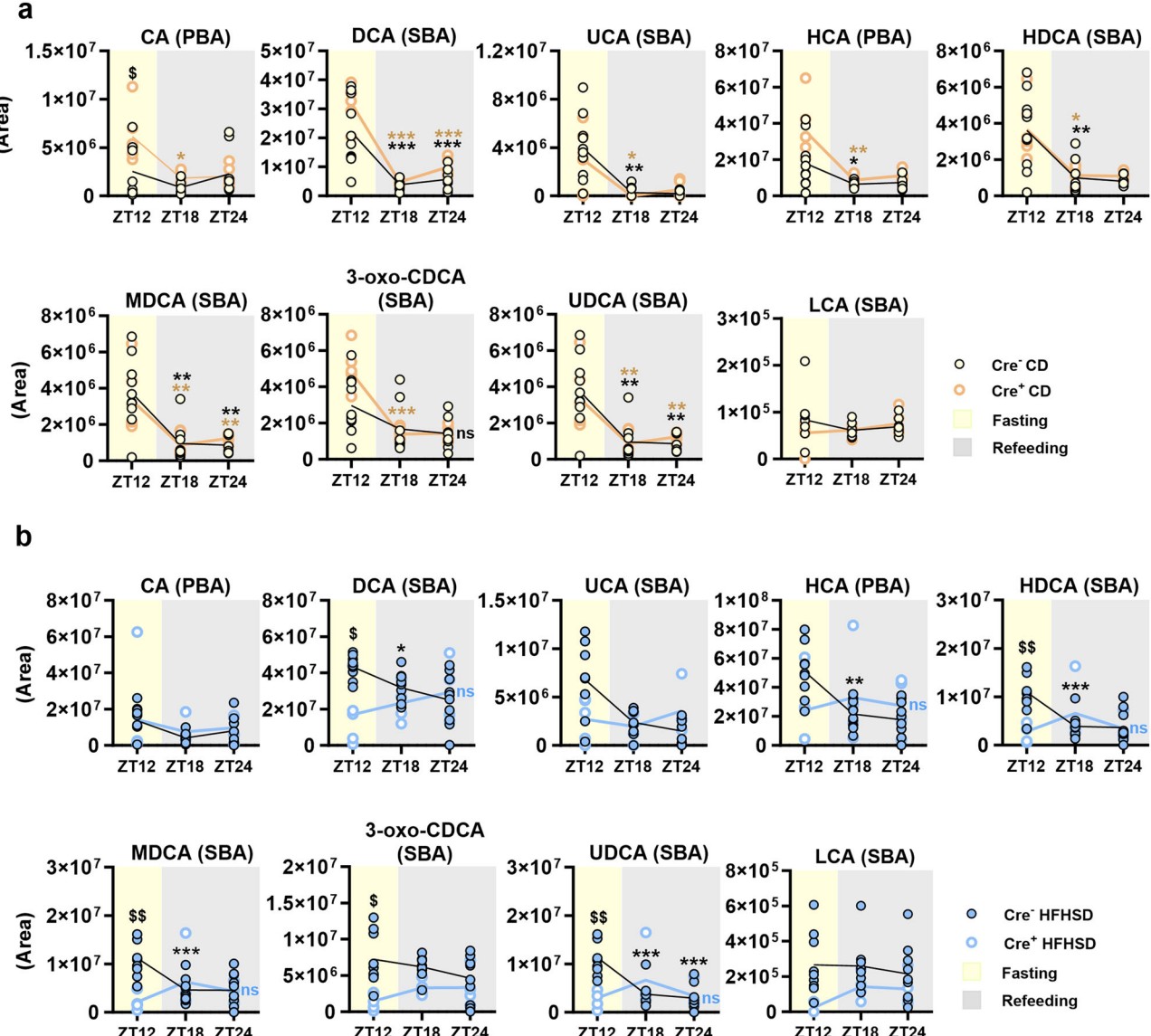

**Fig. 4 | Influence of the ablation of Nav1.8+ neurons on fecal bile acids levels during dark-restricted feeding. a** From left to right: Fecal levels of the primary bile acid (PBA) cholic acid (CA), and its derived secondary bile acids (SBAs) deoxycholic acid (DCA) and ursocholic acid (UCA); Fecal levels of the PBA hyocholic acid (HCA), and its derived SBAs hyodeoxycholic acid (HDCA) and murideoxycholic acid (MDCA); Fecal levels of the SBAs 3-oxo-chenodeoxycholic acid (3-oxo-CDCA), ursodeoxycholic acid (UDCA) and lithocholic acid (LCA), derived from chenodeoxycholic acid (the latter not shown) in control and Nav1.8-cre/DTA mice at the end of the dark-restricted feeding of control diet (CD) after 12 h of fasting (zeitgeber time, ZT12), and 6 h and 12 h of refeeding (ZT18 and ZT24, respectively). **b** The metabolites mentioned in (**a**) measured in the same ZTs in control and Nav1.8-cre/DTA mice fed high-fat high-sugar diet (HFHSD) at the end of the dark-restricted feeding. Data are represented as scatter plots indicating individual values and the mean follow-up curve shown as chromatographic peak area. (Cre⁻ CD/HFHSD, $n = 9$–10 mice and Cre⁺ CD/HFHSD, $n = 4$–6 mice). Mice fed CD are depicted by solid yellow (Cre⁻) or empty orange (Cre⁺) circles, mice fed HFHSD are depicted by solid blue (Cre⁻) or empty blue (Cre⁺) circles; mean values in the follow-up curves are represented by black (Cre⁻ CD, Cre⁻ HFHSD), bold orange (Cre⁺ CD) or bold blue (Cre⁺ HFHSD) lines; shadings in light yellow and grey represent fasting (light phase) and refeeding (dark phase) periods, respectively. The univariate *t*-test was applied to study separately the variations of bile acids according to the genotype (Cre⁻ and Cre⁺) and prandial condition (ZT). $^{\$}p < 0.05$ and $^{\$\$}p < 0.01$ Cre⁻ HFHSD vs Cre⁺ HFHSD or Cre⁻ CD vs Cre⁺ CD for a specific ZT; $*p < 0.05$, $**p < 0.01$ and $***p < 0.001$ vs ZT12.

normal weight gain in response to HFHSD feeding probably by modulating metabolic routes of fat utilisation (Fig. 8).

The ablation of Nav1.8+ neurons dysregulated the control of body weight and food intake. Specifically, under *ad libitum* regimen, Nav1.8-cre/DTA mice were resistant to weight gain in response to HFHSD, and showed higher body weight loss in the resting phase when fed CD. Under food restriction regime, Nav1.8-cre/DTA also showed a disturbed body weight control irrespective of the type of diet with exacerbated body weight loss and gain in fasting-refeeding cycles only under DRF (the active phase of mice).

On the other hand, the absence of these sensory neurons exacerbated food intake in *ad libitum* feeding and DRF, which could result from the depletion of IGLEs since they express Nav1.8+ and are involved in mechanical-induced meal-termination[6,8]. Specifically, this effect was observed at the beginning of the dark phase, when food intake is mainly governed by hunger. For instance, in the ZT12-ZT18 period, DRF caused hypophagia after fasting in HFHSD-fed control mice but induced hyperphagia in mice lacking Nav1.8+ neurons pointing out a critical role of these neurons mediating hunger signaling. In addition, Nav1.8+ neurons appear to be involved in nutrient-induced satiation since Nav1.8-cre/DTA mice showed defective Ensure®-induced food intake suppression. The absence of effects under LRF suggests that the role of Nav1.8+ neurons in regulating energy homeostasis after fasting is dependent on light-entrainable cues

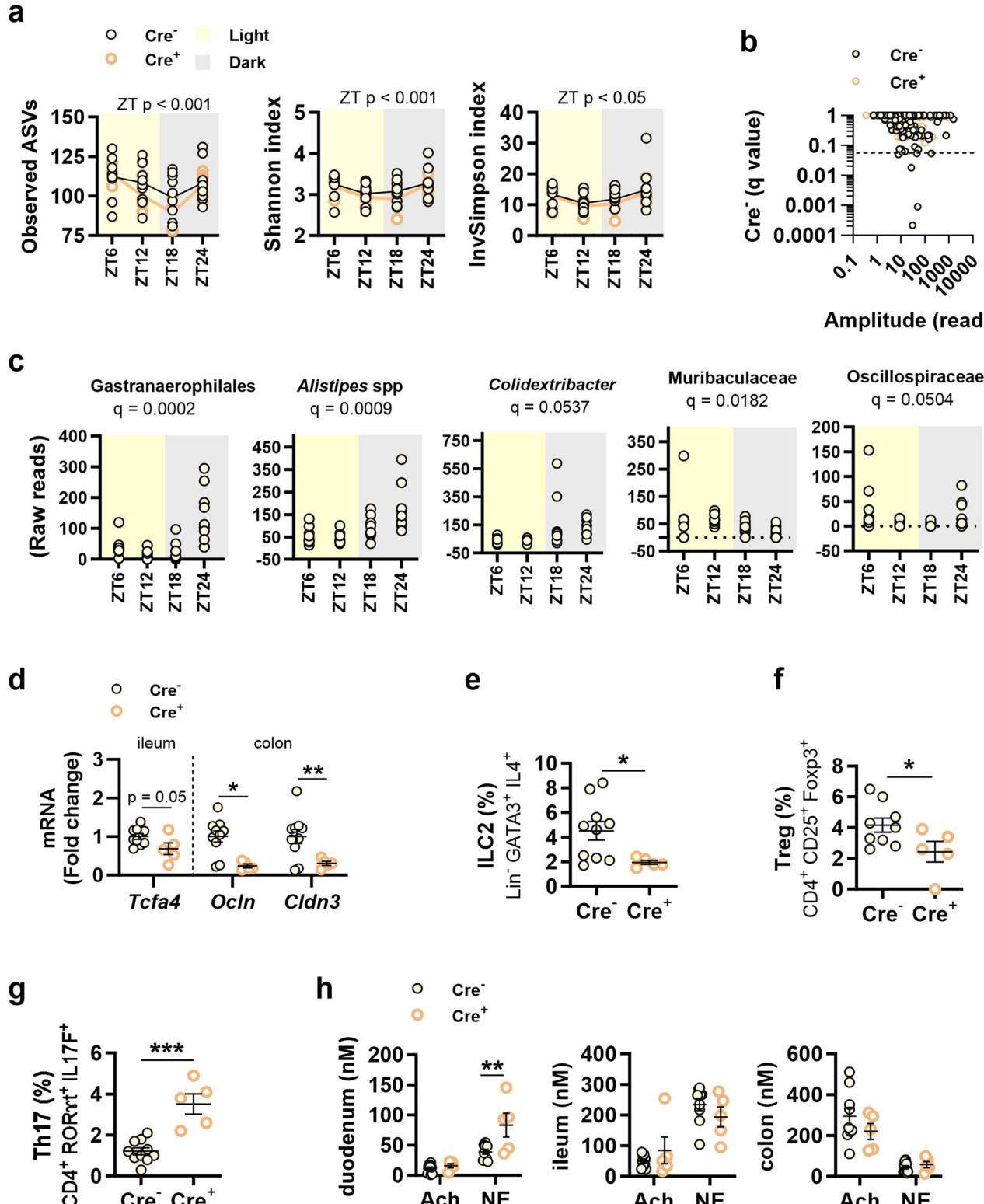

rather than food-entrainable cues. Our findings contrast with the study of Udit et al. that describes a minor role for Nav1.8+ neurons in energy intake in a mouse model of primary afferent neuron ablation[10]. Discrepancies between studies require further investigations, but methodological differences such as the ageing of the animals and/or fasting-refeeding cycles during the food restriction regimen could be involved.

Our data also support an uncoordinated regulation of weight gain and feeding behavior induced by the ablation of Nav1.8+ neurons, since Nav1.8-

cre/DTA mice chronically fed with HFHSD showed attenuated weight gain despite their tendency toward increased food intake. Exploration of possible mechanisms involved in this uncoordinated control suggested that ablation of Nav1.8+ neurons might overstimulate energy-dissipating metabolic pathways, which in turn would contribute to limit fat depots and exacerbate food intake, as a mechanism to compensate for the energy deficient state. In fact, we showed that Nav1.8-cre/DTA mice fed *ad libitum* HFHSD tended to have higher UCP1 expression in BAT, suggesting increased thermogenic

**Fig. 5 | Effects of Nav1.8+ neuron ablation on the daily variations of gut microbiota and on gut barrier integrity and defence in mice fed control diet *ad libitum*.** At the end of the experiment, in Nav1.8-cre/DTA mice (Cre⁺) and their control littermates (Cre⁻) fed control diet (CD), we measured: **a** Alpha diversity of fecal microbiota every 6 h for 24 h, measured as observed amplicon sequence variants (ASVs) and the Shannon and Inverse Simpson indices. **b** Amplitude of ASVs daily oscillations. The dashed line indicates a significant cut-off of $p \leq 0.05$. **c** Representation of the abundance of oscillating ASVs throughout the day; **d** mRNA levels of transcription factor 4 (*Tcf4*), occludin (*Ocln*) and claudin 3 (*Cldn3*) in ileum and colon. **e–g** Percentage of type 2 innate lymphoid cells (ILC2) (**e**), regulatory T cells (Treg) (**f**) and T helper 17 cells (Th17) (**g**) in the lamina propria of the small intestine. **h** Acetylcholine (Ach) and noradrenaline (NE) levels in duodenum, ileum and colon. Data are represented as scatter plots for individual values with mean follow-up curve or as scatter plots indicating individual values ± SEM (Cre⁻ $n = 9$–10 mice and Cre⁺ $n = 5$ mice). Mice fed CD are depicted by solid yellow (Cre⁻) or empty orange (Cre⁺) circles; mean values in the follow-up curves in (**a**) are represented by black (Cre⁻) or bold orange (Cre⁺) lines; shadings in light yellow and grey represent light and dark phases, respectively. **a:** Two-way ANOVA with genotype (Cre⁻ or Cre⁺) and zeitgeber time (ZT) as between-subject factors followed by Bonferroni's *post hoc* test. **b:** JTK approach with Benjamini and Hochberg multiple testing correction (DiscoRhythm); **d–h:** unpaired Student's *t* test; *$p < 0.05$ and **$p < 0.01$.

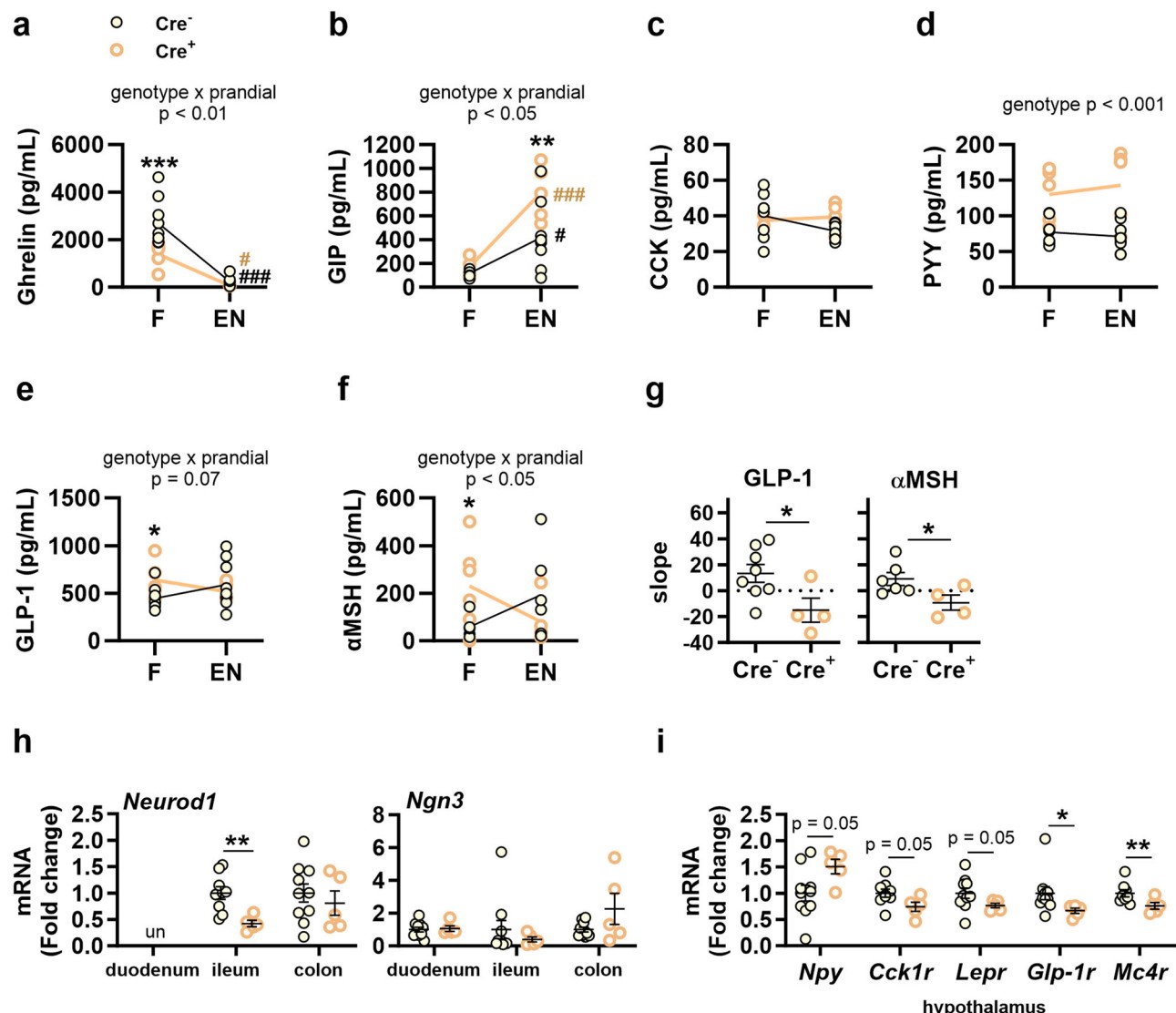

**Fig. 6 | Effects of the ablation of Nav1.8+ neurons on gut-brain orexigenic and anorexigenic signals in mice fed control diet *ad libitum*.** At the end of the experiment in Nav1.8-cre/DTA mice (Cre⁺) and their control littermates (Cre⁻) fed control diet (CD), we measured the following: **a–f** In 4-h-fasted mice (F) and 15 min after an oral load of Ensure® (EN), plasma levels of **a** Ghrelin, **b** GIP, **c** CKK, **d** PYY, **e** GLP-1 and **f** αMSH. **g** Slope calculated between fasting and post-Ensure® levels of GLP-1 and αMSH. **h** mRNA levels of *NeuroD1* and *Ngn3* in duodenum, ileum and colon. **i** mRNA levels of *Npy*, *Cck1r*, *Lepr*, *Glp1r* and *Mc4r* in the hypothalamus. Data are represented as scatter plots for individual values with mean follow-up curve or as scatter plots indicating individual values ± SEM (Cre⁻ $n = 9$–10 mice and Cre⁺ $n = 5$ mice). Mice fed CD are depicted by solid yellow (Cre⁻) or empty orange (Cre⁺) circles; mean values in (**a–f**) are represented by black (Cre⁻) or bold orange (Cre⁺) lines. **a–f:** Two-way ANOVA with genotype (Cre⁻ or Cre⁺) and prandial condition (F/EN) as between-subject factors (main effects and interactions are indicated in the top of the graphs) followed by Bonferroni's *post hoc* test (**a–c**) or Student's *t* test (**e**, **f**); **g**, **h**, **i:** unpaired Student's *t* test (for *Glp1r* in (**i**) Mann–Whitney *U* test was conducted). *$p < 0.05$, **$p < 0.01$, ***$p < 0.001$ vs Cre⁻ (when applicable in a specified prandial condition) and #$p < 0.05$, ###$p < 0.001$ vs fasting condition for a specified genotype (black for Cre⁻ and orange for Cre⁺).

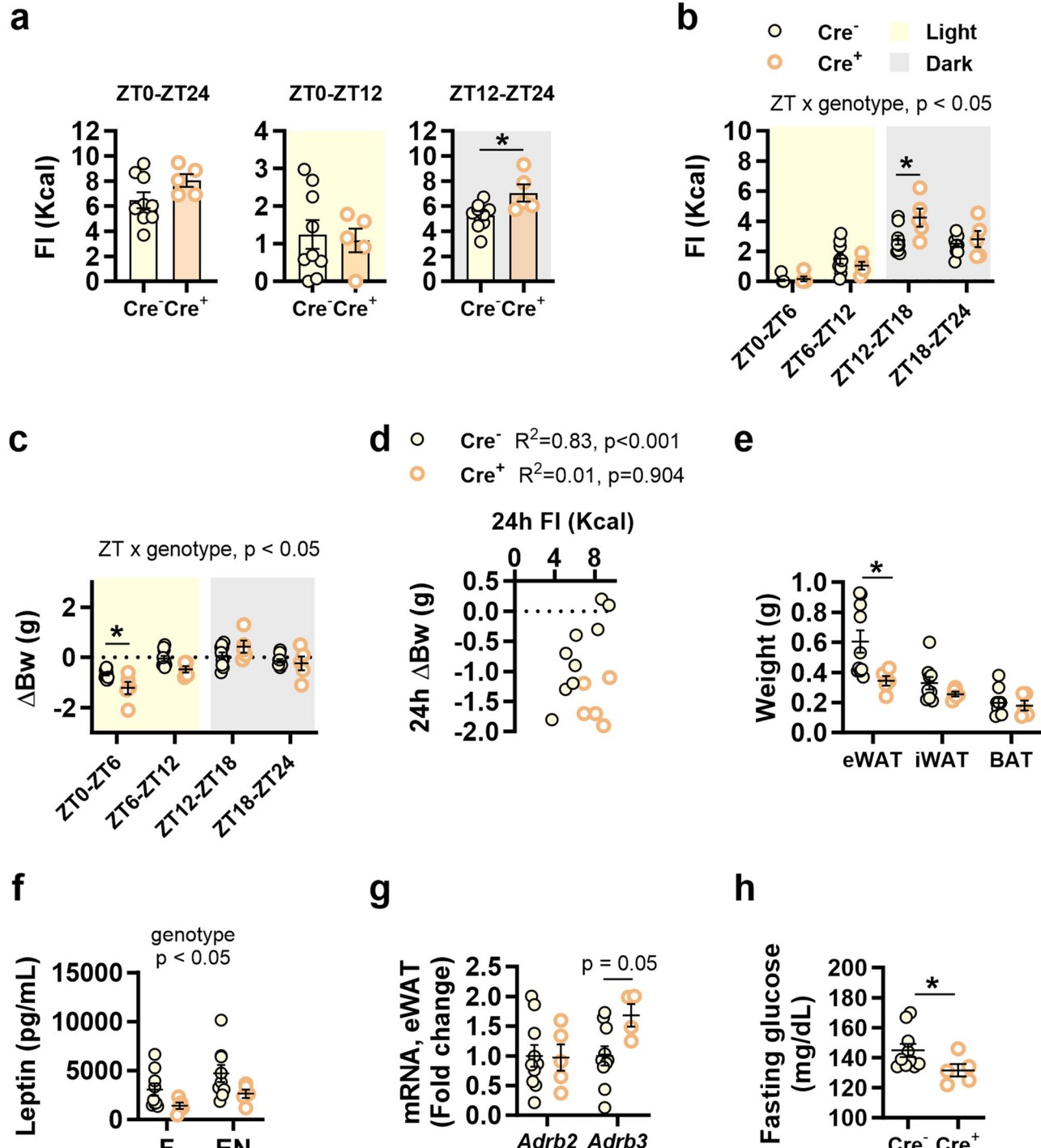

**Fig. 7 | Impact of the ablation of Nav1.8+ neurons on daily food intake and body weight in mice fed control diet *ad libitum*.** At the end of the experiment we measured: **a** *Ad libitum* food intake (FI) of control diet (CD) in Nav1.8-cre/DTA mice (Cre⁺) and their control littermates (Cre⁻) during 24 h (zeitgeber time, ZT0-ZT24) and in the light and dark phase (ZT0-ZT12 and ZT12-ZT24, respectively). **b** *Ad libitum* FI of CD during ZT0 to ZT6, ZT6 to ZT12, ZT12 to ZT18, and ZT18 to ZT24; **c** Body weight variations (ΔBw) in the same ZT periods as in (**b**). **d** Correlation between 24-h *ad libitum* FI of CD and ΔBw. **e** Weight of different fractions of the adipose tissue: epididymal and inguinal white adipose tissue (eWAT and iWAT, respectively), and brown adipose tissue (BAT). **f** Plasma levels of leptin after 4 h of fasting (F) and after oral administration of Ensure® (EN). **g** Relative gene expression of beta 2 and 3 adrenergic beta receptor genes (*Adrb2* and *Adrb3*) in eWAT. **h** Fasting glycemia. Data are represented by scatter plots indicating individual values with mean ± SEM, (Cre⁻ *n* = 9–10 mice and Cre⁺ *n* = 5–6 mice). Mice fed CD are depicted by solid yellow (Cre⁻) or empty orange (Cre⁺) circles; shadings in light yellow and grey represent light and dark phases, respectively. **b, c, f**: Two-way ANOVA with genotype (Cre⁻ or Cre⁺) and ZT or genotype and prandial condition (F/EN) as between-subject factors (main effects and interactions are indicated in the top of the graphs) followed by Bonferroni's *post hoc* test. **d**: Pearson correlation. **a, e, g**: Student's *t* test; **h**: Mann–Whitney *U* test. *$p < 0.05$.

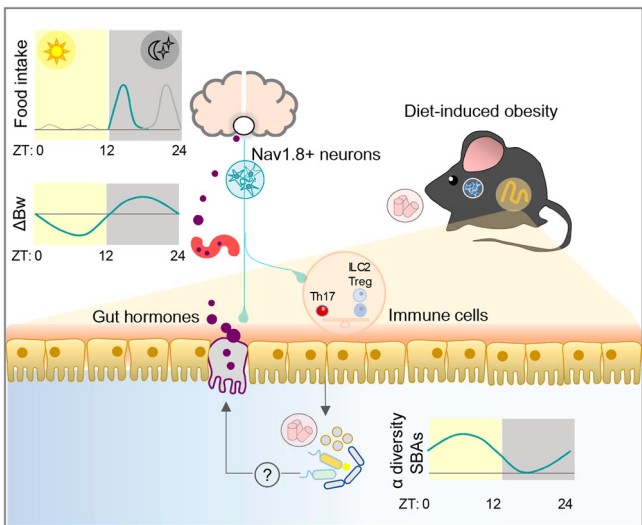

**Fig. 8 | Graphical summary of the main roles of Nav1.8+ neurons on daily control of food intake, body weight and intestinal endocrine, immune and microbial signals.** In mice, Nav1.8+ neurons show a relevant role in regulating food intake at the beginning of the active phase (dusk peak). Body weight variations along the day are also affected by Nav1.8+ neurons since limit body weight loss in the resting phase and after 12 h of fasting. At intestinal level, these neurons maintain the homeostasis of immune system, shape pre- and postprandial gut microbiota oscillations, and preserve the secretion of gut hormones that transmit hunger and satiety. At long-term, Nav1.8+ neurons are required to develop diet-induced obesity. ZT zeitgeber time.

potential helping to limit their fat depots. In line with this, previous investigations revealed that in response to Western diet feeding, mice without the lipid sensing nuclear receptor LXRα/β (liver X receptor alpha and beta) in Nav1.8-expressing neurons also showed increased thermogenesis activity in BAT coupled with reduced body weight gain[22]. Further supporting an increased energy loss in Nav1.8-cre/DTA mice, our study also identified reduced eWAT depots in these mice fed *ad libitum* CD accompanied by the overexpression of the β3 adrenergic receptor, suggesting higher sympathetic-mediated fatty acid release for oxidation[23]. In line with this, it was recently demonstrated that the ablation of Nav1.8+ neurons in mice reduces food efficiency and lipid absorption after a meal[24] suggesting a limited contribution of HFHSD to weight gain in this animal model.

On the other hand, we observed that Nav1.8-cre/DTA mice did not survive longer periods of HFHSD consumption (14 weeks in cohort 2-related experiments). Nav1.8+ sensory neurons have been shown to protect host from endotoxic death[25] and to be necessary for immune homeostasis in response to an excess of dietary lipids[10]. Accordingly, chronic HFHSD feeding could trigger inflammatory processes in Nav1.8-cre/DTA mice, underlying the enhanced mobilization of lipids that is seen in cachexia[26] and ultimately causing death due to their inability to tolerate the exacerbated lipid-induced inflammation.

Thus, due to the inability of Nav1.8-cre/DTA to survive 14 weeks of HFHSD feeding, the impact of the ablation of Nav1.8+ neurons in the gut-brain circuits controlling food intake was explored only in mice fed with CD. In line with the increased food intake, we identified an upregulated and downregulated orexigenic and anorexigenic signaling in the hypothalamus, respectively. Nav1.8+ neurons appear to control food intake by maintaining the secretion of orexigenic and anorexigenic gut hormones, especially ghrelin, GLP-1 and PYY, during the pre- and postprandial periods. Intestinal sympathetic and parasympathetic innervations control ghrelin and GLP-1 secretion, respectively[27–29]. Accordingly, changes in efferent tone resulting from the absence of Nav1.8+ afferents could potentially affect gut hormone secretion. In this line, duodenal noradrenaline levels were increased in Nav1.8-cre/DTA mice, and thus sympathetic-mediated stimulation of cells producing ghrelin could initially increase ghrelin levels

followed by long-term aberrant ghrelin secretion due the chronic over-stimulation, resulting in reduced fasting levels of this hormone. The diminished levels of ghrelin in fasting Nav1.8-cre/DTA mice were not sufficient to normalize food intake, as the postprandial enteroendocrine signaling inducing meal termination was also dysregulated. For instance, the ablation of Nav1.8+ neurons induced the oversecretion of GLP-1 during fasting and a lack of response after a meal. Studies in vagotomized rats propose a proximal-distal neuroendocrine loop underlying the early post-prandial secretion of GLP-1. This signaling is initiated in the proximal small intestine by nutrient-dependent stimulation of vagal afferents to indirectly induce GLP-1 secretion in the distal gut through parasympathetic innervations[28]. Our study suggests that Nav1.8+ neurons might target this loop, explaining that loss of these neurons could contribute to disturbed L-cell functionality, affecting also PYY secretion. On the other hand, Nav1.8+ neurons are required to maintain plasma αMSH levels, which diffuse from the CNS to circulation[30]. Mice lacking Nav1.8+ neurons showed increased fasting levels of αMSH, which could overstimulate PYY and GLP-1 secretion through MC4R in L-cells[31].

We also observed an enhanced proinflammatory tone in the intestinal mucosa of mice lacking Nav1.8+ neurons, which could contribute to the observed long-term dysregulation of energy homeostasis. The proportions of ILC2 and Treg cells were reduced in the lamina propria of the small intestine in Nav1.8-cre/DTA mice, coupled to an increase in Th17 cell number. The loss of Nav1.8+ neurons appears to alter intestinal immunity by disrupting neuro-immune communication mediated by Nav1.8+ neurons and/or the efferent sympathetic and parasympathetic innervations in the gut. In the context of neuro-immune communication, the immuno-modulatory neuropeptide VIP was almost undetectable in the absence of Nav1.8+ neurons, which produce this neuropeptide[21]. Thus, loss of Nav1.8+ neurons would interrupt VIP-ILC2 communication, dysregulating the intestinal type 2 inflammatory response as occurs in the pulmonary mucosa[21]. Correspondingly, as Nav1.8+ neurons express immune-related receptors (i.e., for prostaglandins and leukotrienes)[8], their absence would change the anti-inflammatory vagovagal reflex, and/or the intestinal sympathetic tone. The sympathetic tone acts through the β2-adrenergic receptor to maintain ILC2 cell functionality, allowing acute protective type 2 responses but preventing pathological ILC2 overstimulation[32]. In this regard, the increased noradrenaline levels in the upper gut of Nav1.8-cre/DTA mice suggest that an upregulated sympathetic tone could contribute to reducing the abundance of ILC2 cells. Indeed, in bacterially-infected mice, β2-adrenergic receptor agonists inhibit intestinal ILC2 cell proliferation, resulting in higher worm burdens due to the attenuated type 2 response[32]. On the other hand, the reduced abundance of Treg cells in Nav1.8-cre/DTA mice could be driven by the higher intestinal noradrenaline content, related to a sympathetic-mediated apoptotic effect on these cells[33,34]. Therefore, the disrupted intestinal immune homeostasis associated with the loss of Nav1.8+ neurons could prompt systemic inflammation, suggested by the observed splenomegaly. This could, in turn, contribute to dysregulating energy balance, leading to reduced energy depots despite the enhanced food intake. Indeed, Nav1.8-cre/DTA mice showed changes in plasma metabolic markers suggestive of energy deficiency (reduced leptin and glucose levels).

The gut microbiota and its interactions with diet, which ultimately modulate gut-to brain neuroendocrine signaling, could also be affected by the greater vulnerability to inflammatory dietary signals through the loss of Nav1.8+ neurons.

The alpha diversity of fecal gut microbiota was not significantly affected by the loss of Nav1.8+ neurons under CD feeding *ad libitum*, in line with other investigations[9]. Nevertheless, their loss impeded the daily rhythms of *Alistipes* spp. and *Colidextribacter* spp., whose amplitudes in control mice were increased during feeding periods. This suggests that nutrient supply in the intestinal lumen contributes to the increased postprandial abundance of these genera. Indeed, *Alistipes finegoldii* assembles its membrane lipids from dietary fatty acids[35], which are more available after feeding.

Under DRF, we could more acutely identify the pre- and postprandial variations of the gut microbiota and its associated metabolites dependent on

https://doi.org/10.1038/s42003-024-05905-3 **Article**

Nav1.8+ neurons. The gut microbiota structure and derived BAs were particularly affected by the loss of Nav1.8+ neurons under HFHSD. Indeed, their loss impeded the refeeding-induced decline in gut microbiota diversity. The postprandial reduction of alpha diversity in control mice suggests an increase in the abundance of some microbial taxa and a reduction in others in response to fasting-induced refeeding. Although we could not detect major interactions between the genotype and the zeitgeber time on specific ASVs, we found an impaired fasting-to-refeeding decrease in *Alistipes* spp. Tentatively, based on previous investigations[9], greater prandial-related ASV variations dependent on the genotype could be detected in the ileal intestinal content.

The loss of Nav1.8+ neurons also blunted the reduction in fecal SBA levels induced by refeeding due to their reduced levels under fasting. This could be the result of lower hepatic PBA production, lower SBA production (owing to compositional microbial changes) or to higher intestinal absorption. An imbalance in the production of BAs by hepatocytes under HFHSD is a likely possibility, being continuous and coordinated in control mice in response to lipid intake but discontinuous and uncoordinated in Nav1.8-cre/DTA mice, reflecting their dysregulated feeding patterns. Correspondingly, the results also indicate that the gut microbiota of control mice can convert unused PBAs into SBAs upon reaching the colon in the fasting period instead of being absorbed. Contrastingly, the reduced production of SBAs of fasted mice lacking Nav1.8+ neurons could be explained by the absence of PBAs reaching the colon, as they were completely used in the upper gut through the digestion of a higher amount of lipids because of the evident hyperphagia. Also, the changes of the fasted gut microbiota composition of Nav1.8-cre/DTA mice might underlie their inability to produce SBAs from PBAs.

Recent studies have revealed that BAs modulate feeding behavior and energy metabolism by favoring the secretion of gut hormones[36], intestinal lipid sensing[37], and hypothalamic TGR5 signaling, to modulate the sympathetic efferent tone of the adipose tissue[38]. Assuming that the altered profile of BA in feces induced by the loss of Nav1.8+ neurons produces similar alterations in the distal intestinal content, the reduction in BAs during fasting could contribute to disrupting enteroendocrine signaling in Nav1.8-cre/DTA mice and, consequently, their daily control of food intake and body weight variations.

In summary, we have identified a role of Nav1.8-expressing neurons in conveying peripheral signals to the brain to govern daily food intake and body weight variations. We propose that these neurons contribute to energy homeostasis not only because they are part of the neural signaling network transmitting peripheral sensory information to the brain, but also because they are necessary to maintain (1) prandial regulation of the enteroendocrine system, (2) fasting body weight loss, and (3) prandial gut microbiota configurations and BA signaling in the distal gut. Accordingly, the loss of Nav1.8+ neurons could account for the overconsumption, inflammation and dysbiosis in obesity, and for the exacerbated weight loss and inflammation in cachexia[39]. Further studies are needed to determine the spinal or vagal nature of the Nav1.8+ neurons controlling food intake, body weight and endocrine, immune and microbial intestinal signals, as well as, the possible causal role of the microbial changes identified on the gut-brain axis for controlling food intake and energy homeostasis.

## Methods
### Generation and validation of Nav1.8-cre/DTA mice
Heterozygous Nav1.8 knock-in Cre-recombinase male mice (EM:04582, EMMA-infrafrontier repository, Munich, Germany) were crossed with homozygous ROSA26-eGFP-DTA female mice (stock #006331; The Jackson Laboratory) to obtain 1:1 litters of Nav1.8-deficient mice (mice expressing Cre and DTA alleles: Nav1.8-cre/DTA, indicated as Cre+ in Fig.s) and control mice (mice with DTA but without Cre allele: control littermates, indicated as Cre− in Fig.s). Mice were bred in barrier facility using ventilated racks and cages at the animal facility of the University of Valencia (Animal Production Section, SCSIE, University of Valencia, Spain). Only male offspring were employed in the present study, females were employed for other

investigations. Genomic DNA from ear punches of male mice was used for PCR; wild-type or Nav1.8-Cre fragments and wild-type and floxed DTA fragments were amplified in an Eppendorf thermocycler using the Phusion High-Fidelity PCR Kit (Thermo Fisher Scientific) and primer sequences detailed in Supplementary Table 1. For the mSNS13s/mSNS12a primer pair PCR, DNA was incubated at 94 °C for 2 min followed by 34 cycles of denaturation at 94 °C for 30 s, annealing at 63 °C for 45 s and extension at 72 °C for 1 min, with a final step at 72 °C for 10 min. For the mSNS13s/Cre5a and Cre2s/Cre5a primer pair PCR, DNA was incubated at 94 °C for 2 min followed by 29 cycles of denaturation at 94 °C for 30 s, annealing at 62 °C for 30 s and extension at 72 °C for 1 min, with a final step at 72 °C for 10 min. For the ROSA26 F/floxed DTA and ROSA26 F/ROSA 26 R primer pair PCR, DNA was incubated at 98 °C for 3 min followed by 30 cycles of denaturation at 95 °C for 20 s, annealing at 60 °C for 45 s and extension at 72 °C for 20 s, with a final step at 72 °C for 5 min. The amplified fragments were separated by electrophoresis.

The effectiveness of the Nav1.8 genetic ablation was explored as described previously[10]. In brief, RNA from randomly selected nodose ganglia samples was isolated and reverse transcribed with the Ambion® RiboPure Kit (Invitrogen) and SuperScript™ VILO™ Master Mix (Thermo Fisher Scientific), respectively. To analyze the expression of *Scn10a* (coding for Nav1.8), cDNA was first pre-amplified in triplicate with the TaqMan® PreAmp Master Mix (Applied Biosystems, Thermo Fisher Scientific) and incubated at 95 °C for 10 min followed by 10 cycles of denaturation at 95 °C for 15 s and annealing/extension at 60 °C for 4 min, with a final step for enzyme inactivation at 99 °C for 10 min in an Eppendorf thermocycler. Pre-amplified cDNA was further amplified by qPCR reactions using TaqMan® Gene Expression Assays (FAM-MGB; Applied Biosystems, Thermo Fisher Scientific; references provided in Supplementary Table 1) and incubated with TaqMan® Fast Advanced Master Mix (Applied Biosystems, Thermo Fisher Scientific) at 50 °C for 2 min and at 95 °C for 20 s, followed by 40 cycles of denaturation at 95 °C for 3 s and annealing/extension at 60 °C for 30 s in a LightCycler® 480 Instrument (Roche). *Rpl19* was used as a housekeeping gene. The $2^{-(\Delta\Delta Ct)}$ method was employed to calculate relative gene expression, represented as fold change relative to Cre− mice for each gene.

### Mouse experimental design and metabolic phenotyping
Two cohort of male mice were generated to conduct the experiments described in this study. Supplementary Fig. 1 details the experimental procedure. For all experiments, six-week-old male mice were individually housed and maintained under constant conditions of humidity and temperature (23 ± 2 °C) and a regular 12-h light-dark cycle, in which zeitgeber time (ZT)0 corresponds to lights on and ZT12, to lights off.

For experiments using cohort 1 (Supplementary Fig. 1a), Nav1.8-ablated mice ($n = 4$) and their control littermates ($n = 6$) were fed *ad libitum* with HFHSD (D12451; 45% of energy from fat and 21% from sucrose, Research Diets) for 8 weeks. Body weight was weekly determined and 24 h-food intake was measured at the end of the experiment. Mice were anaesthetized with an intraperitoneal injection of pentobarbital (50 mg/kg) and then transcardially perfused with PBS followed by 10% neutral buffered formalin. BAT was isolated for immunohistochemistry.

For experiments using cohort 2 (Supplementary Fig. 1b), Nav1.8-ablated mice ($n = 13$) and their control littermates ($n = 20$) were fed either CD (D12450K containing 10% of energy from fat and without sucrose, Research Diets) or HFHSD for 14 weeks.

Throughout this period, animals were submitted to the following feeding regimes: 5 weeks of free access to food: *ad libitum* feeding (week 0–5); followed by 3 weeks of free access to food only during the light and the dark phase sequentially: light- (week 6-8) and dark- (week 9–11) restricted feeding, respectively (LFR and DRF). During the period of *ad libitum* feeding, body weight was monitored weekly. In both LRF and DRF, body weight loss and body weight gain were calculated daily after 12 h of fasting and 12 h of refeeding, respectively. Under these feeding schedules, food intake was also determined at 6 h and at 12 h of the refeeding period for

2 consecutive days, and the feeding response to an oral load (nutritional challenge) of a mixed nutrient solution (Ensure®) was explored. In parallel, feces were collected after 12 h of fasting, and 6 h and 12 h of refeeding. After the food restriction period, mice were switched again to an *ad libitum* feeding schedule for 3 weeks (week 12–14). In the last 24 h of this period, food intake and body weight variations were determined every 6 h and fecal samples were collected. At the end of the experiment (14 weeks), the secretion of energy metabolism-related hormones in response to the oral administration of Ensure® (detailed in "feeding response to oral administration of Ensure" subsection) was determined. Mice were sacrificed by cervical dislocation for sample collection. Adipose tissues (eWAT, iWAT and BAT) and spleens were weighed. Brain, adipose tissue and the first and the last 2-cm of the upper and distal small intestine (duodenum and ileum, respectively) and colon were immediately frozen and stored at −80 °C. The remaining small intestines and the spleens were embedded in cold FACS buffer for further isolation of immune cells (detailed in "immune cell isolation and immune phenotyping" subsections).

We have complied with all relevant ethical regulations for animal use. All experimental procedures using animals were in accordance with European Union 2010/63/UE and Spanish RD53/2013 guidelines and approved by the ethics committee of the University of Valencia (Animal Production Section, SCSIE, University of Valencia, Spain) and authorized by Dirección General de Agricultura, Ganadería y Pesca (Generalitat Valenciana) (approval ID 2019/VSC/PEA/0020 and 2020/VSC/PEA/0022).

### Feeding response to oral administration of Ensure®

12-h fasted mice received by oral gavage saline solution or Ensure® (nutritional challenge) (12.4 kcal/kg body weight Ensure®, Abbot, or the corresponding volume of saline solution) at ZT0, in the LRF schedule or ZT12, in DRF. After 10 min, mice had free access to food to determine food intake after 120 min from when food was provided. Saline trials were conducted before the Ensure® trials for four consecutive days, which was the time required to obtain a regular feeding response. A similar number of trials were performed with Ensure®. Ensure®-induced food intake suppression was calculated as the percentage of food intake suppression after an oral gavage of Ensure® relative to food intake after oral gavage of saline.

### In vivo hormone secretion assay

Blood from the submandibular vein was collected after 4 h of fasting and 15 min after an oral gavage of Ensure® (12.4 kcal/kg of body weight) in Microvette® 500 K3E tubes (Sarstedt) containing inhibitors of DPPIV and serine proteases (Pefabloc) (Sigma). Blood was immediately centrifuged and plasma was stored at −80 °C until hormone measurement.

### Glycemia and hormone measurements

Before blood centrifugation, glycemia was measured using glucose test strips and a glucometer (Contour® Next meter). The levels of glucagon, active ghrelin, insulin, leptin, total PYY, total GIP and total GLP-1 were measured using a MILLIPLEX Mouse Metabolic Hormone Expanded Panel-7 Plex on a Luminex® MAGPIX System (Milliplex, Merck Group). CCK and αMSH were quantified in plasma using Mouse Cholecystokinin 8 (CCK8) ELISA kit (MyBioSource, Quimigen) and Mouse αMSH (Alpha-Melanocyte Stimulating Hormone) ELISA Kit (Elabscience, Quimigen), respectively, on a CLARIOstar Microplate reader (BMG Labtech).

### Immunohistochemistry

UCP1 immunostaining was conducted in BAT samples according to standardized protocols by Patologika Laboratorio S.L. (Valencia, Spain). Images were obtained using an Eclipse 90i Nikon wide-field microscope (Nikon Corp., Tokyo, Japan) with a CFI Plan Fluor DIC M/N2 (MRH00200) Nikon dry-air objective (20× or 4×) combined with an optical zoom factor of 0.8×. Fiji software (ImageJ 1.49q Software, National Institutes of Health) and Nis elements BR 3.2 software (Nikon Corp.) were used for quantification according to standardized protocols of the imaging service from IATA-CSIC.

### Immune cell isolation from spleen and the small intestinal epithelium and lamina propria

Spleens were gently crushed using the plunger of a syringe and washed with FACS buffer (PBS-10% fetal bovine serum, Thermo Scientific). The splenic suspensions were filtered (70 μm nylon cell strainers, Biologix) and centrifuged (450 g, 5 min, 4 °C). Isolated cells were preserved in cold FACS buffer until immunolabeling.

Small pieces of longitudinally-opened small intestines were incubated twice with calcium/magnesium-supplemented Hank´s balanced salt solution (HBSS) (Thermo Fisher Scientific) with 5 mM EDTA, 1 mM DTT and antibiotics (100 μg/mL streptomycin-100 U/mL penicillin, Merck) for 30 min at 37 °C. After each incubation, intestinal solutions were filtered (100-μm nylon cell strainers, Biologix) and centrifuged to collect epithelial cells. Cells from the lamina propria were isolated by incubating the remaining intestinal tissues twice with HBSS containing antibiotics and 0.5 mg/mL collagenase D, 1 mg/mL DNase I (Roche), and 3 mg/mL dispase II (Sigma) for 30 min at 37 °C. Cell suspensions were filtered (70-μm nylon cell strainers, Biologix) and centrifuged (450 g, 5 min, 4 °C) to collect lamina propria cells. Cells from both intestinal compartments were preserved in FACS solution until immunolabeling.

### Immune phenotyping by flow cytometry

The isolated cells from spleen, intestinal epithelium and intestinal lamina propria were incubated with a combination of flow cytometry antibodies against surface and intracellular immune markers for 30 min at 4 °C (protected from light). For intracellular marker immunolabeling, cells were first permeabilized and fixed using a fixation/permeabilization solution (BD Bioscience). The combinations of anti-mouse flow cytometry antibodies for labeling specific immune cell populations are indicated in Supplementary Table 2 and Supplementary Fig. 7. Spleens were tested for macrophages (M1 and M2) and CD4 + T cells, including memory and effector CD4 + T cells. The intestinal epithelium was tested for ILC, including NK and ILC1. The lamina propria was tested for macrophages (M1 and M2), ILC2 and ILC3, Treg and Th17 cells.

Data acquisition and analysis were performed using a BD LSRFortessa flow cytometer operated with FACS Diva software v.7.0 (BD Biosciences).

### RNA isolation and RT-qPCR

Total RNA from hypothalamus, eWAT, duodenum, ileum and colon was isolated using TRIsure™ reagent (Bioline). 1–2 μg of total RNA was retrotranscribed with high-capacity cDNA reverse transcription kit (Applied Biosystems, Thermo Fisher Scientific) incubated at 25 °C (10 min), 37 °C (120 min) 85 °C (5 min) in an Eppendorf thermocycler. Gene amplification was conducted using 300 nM of gene-specific primer pair sequences (Supplementary Table 3) with a cDNA dilution previously validated in our lab and the LightCycler 480 SYBR Green I Master Mix (Roche) on a LightCycler® 480 Instrument (Roche). Changes of gene expression were calculated based on $2^{-\Delta\Delta Ct}$ method and expressed as fold-change expression relative to the control group. Conditions of the qPCR reactions are indicated in Supplementary Table 3.

### Quantification of intestinal acetylcholine, noradrenaline and vasoactive intestinal peptide

For the analyses of Ach and NE, fractions from the upper and distal small intestine (duodenum and ileum, respectively) and colon (~10 mg) were homogenized with 80% acetonitrile with a Tissue Lyser (Qiagen) and centrifuged (9400 g, 10 min, 4 °C) to collect supernatants. NE and Ach in the collected supernatants were measured using a linear ion trap quadrupole LC/MS/MS mass spectrometer (QTRAP ® 6500 LC-MS/MS System, Sciex) according to standardized protocols by SCSIE (University of Valencia). Analytes were separated with a Kinetex C8 column (1.7 μm, 2.1 × 100 mm, Phenomenex) using formic acid (0.1%) and acetonitrile as mobile phases with a 0.35 mL/min flow rate and a gradient elution program. VIP content in the duodenum and ileum was determined using a mouse vasoactive intestinal peptide (VIP) ELISA Kit (Elabscience).

## Bile acid analysis in feces

**Chemicals.** Acetonitrile and water 0.1% (v/v) formic acid were purchased from J.T. Baker (Deventer), and formic acid was obtained from Panreac. Authentic standards of 3,7-Dihydroxy-5-cholan-24-oic Acid (chenodeoxycholic acid) and 3-Hydroxy-11-oxo-5-cholan-24-oic Acid (3-oxo-chenodeoxycholic acid) were purchased from Avanti Polar Lipids (Alabaster, AL, USA). The N-(3α,7α,12α-trihydroxy-5β-cholan-24-oyl)-glycine (glycocholic acid), N-(3α,7α,12α-trihydroxy-5β-cholan-24-oyl)-taurine (taurocholic), 3α,7α,12α-trihydroxy-5β-cholan-24-oic acid (cholic acid), 3α,7β,12α-Trihydroxy-5β-cholan-24-oic acid (ursocholic), 3α,6β-Dihydroxy-5-cholan-24-oic acid (murideoxycholic acid), 3α,6α-Dihydroxy-5β-cholan-24-oic acid (hyodeoxycholic acid), 3α,7β-Dihydroxy-5β-cholan-24-oic acid (ursodeoxycholic acid), 3α,12α-Dihydroxy-5β-cholan-24-oic acid (deoxycholic acid), 3α-Hydroxy-5β-cholan-24-oic acid (lithocholic) and 3β-Hydroxy-5β-cholan-24-oic acid (isolithocholic) were purchased from Cayman Chemical.

## Extraction of bile acids

Bile acids extraction was performed according to the protocol previously described[40] with modifications: 50 µL of 1:10 dilution stool samples were suspended in 450 µL of 50 mM cold sodium acetate buffer (pH 5.6) and were then mixed with 1.5 ml of ethanol and under stirring during 20 min at 25 °C. After centrifugation, the supernatant was diluted four times with water and eluted through C18 cartridge (500 mg/3 ml; Hypersep Spe, Thermo Scientific). The cartridge was washed with ethanol (3 mL) and bile acids were eluted with ethanol (3 mL). Then, the solvent was evaporated and the residue dissolved in 500 µL of methanol. Finally, the solution was filtered by a 0.22 µm PVDF filter (Merck Millipore) prior to the UPLC-ESI-QTOF-MS analysis.

**UPLC-ESI-QTOF-MS analysis.** The metabolomics analysis was performed on a U-HPLC (Infinity 1290; Agilent) coupled to a high-resolution mass spectrometer with a quadrupole time-of-flight mass analyser (6550 iFunnel Q-TOF LC/MS; Agilent) with an Agilent Jet Stream (AJS) electrospray (ESI) source. The mass analyzer was operated in negative mode under the following conditions: gas temperature 150 °C, drying gas 14 L/min, nebulizer pressure 40 psig, sheath gas temperature 350 °C, sheath gas flow 11 L/min, capillary voltage 3500 V, fragmentor voltage 100 V, and octapole radiofrequency voltage 750 V. Data were acquired over the m/z range of 50–1700 at the rate of 2 spectra/s. The m/z range was autocorrected on reference masses 112.9855 and 1033.9881. The MS/MS target product ion spectra were acquired at m/z 100–1100 using a retention time window of 1 min, collision energy of 30 ev and an acquisition rate of 10 spectra/s. The chromatographic analysis was performed with a reversed-phase C18 column (Poroshell 120, 3 × 100 mm, 2.7 µm pore size) at 30 °C, using water + 0.1% formic acid (Phase A) and acetonitrile + 0.1% formic acid (Phase B) as mobile phases with a flow rate of 0.4 mL/min. The gradient started with 50% B, increased in 4 min to 90% B, in 3 min to 99% B, held for 3 min and decreased to the initial conditions during 1 min. The injection volume for all samples was 2 µL. Raw data were processed by Profinder 10.0 software (Version B.10.0, Agilent software metabolomics, Agilent Technologies) through the application of a batch targeted featured extraction based on an in-house database built for bile acids. After data processing, the bile acids identified were analyzed by Mass Hunter Qualitative 10.0 qualitative (Version B.10.0, Agilent software metabolomics, Agilent Technologies).

## Metataxonomic analysis of microbiota

At the end of the last *ad libitum* regimen, fecal samples were collected every 6 h for 24 h (ZT6, ZT12, ZT18 and ZT24); in the last 24 h of the dark-restricted feeding period, samples were collected after 12 h of fasting (ZT12), and after 6 and 12 h of refeeding (ZT18 and ZT24, respectively) (Supplementary Fig. 1b). Samples were immediately snap-frozen in liquid nitrogen and stored at −80 °C. Before DNA extraction, ~30 mg of feces was incubated for 1 h at 37 °C with lysozyme and mutanolysin (Sigma-Aldrich) for

Gram-positive bacterial cell wall lysis. The QIAamp® PowerFecal® Pro DNA kit (Qiagen) was used for isolation of microbial DNA, but instead of using the vortex adapter described in the protocol, samples were homogenized in the provided bead tubes using a Mini-BeadBeater-8 (BioSpec Products) at medium-high speed 3 times for 1 min, with 1 min incubation on ice between bead-beatings. DNA was eluted in 30 µL of nuclease-free water and yields were quantified by fluorimetry with Qubit® (Invitrogen) using the dsDNA HS Assay reagent kit. Concentration of DNA samples ranged from 1 to 50 ng/µL. For metataxonomic analysis, library preparation was performed using Nextera XT v2 Index (Illumina) targeting the V3-V4 region of the 16 S rRNA gene and sequenced on an Illumina® MiSeq platform (2 × 300 bp paired-end reads) at the Genomics Unit of Institute of Parasitology and Biomedicine "López-Neyra" (IPBLN, National Research Council, Granada, Spain).

Quantification of amplicon sequence variants (ASVs) was performed with the DADA2 v.1.24 R package[41]. Raw reads were truncated after 280 bp for forward and 250 bp for reverse reads, and 30 nucleotides were additionally removed at the start of both paired-end reads. Reads were filtered for quality assurance; clean pairs of reads were merged into contig sequences and chimeric sequences were discarded. Taxonomy was assigned by comparing sequences with the SILVA v.138 database[42]. Taxa with 0 reads in 30% of the samples, in at least 3 experimental groups, were removed from subsequent analyses. The alpha diversity was calculated through the estimation of the observed ASVs, and the Shannon and inverse Simpson indices using the Phyloseq v.1.40 R package[43].

## Statistics and reproducibility

Graphs were plotted with GraphPad Prism 9. Statistical analyses were also performed using GraphPad Prism 9, with the exception of the microbiota-related data, which will be detailed later in this section. When data met assumption of normality and equality of variances, determined by conducting Shapiro–Wilk and Bartlett´s tests, respectively, different parametric tests were conducted. A two-way ANOVA was used to analyse the main effects and interactions between genotype (controls: Cre⁻ and Nav1.8-cre/DTA: Cre⁺) and diet (CD or HFHSD), genotype and prandial condition (fasting/Ensure®), or genotype and time (zeitgeber time [ZT]) as between-subject factors. Multiple comparisons using Bonferroni´s *post hoc* test were performed when interactions were detected. The unpaired Student´s *t* test or the Mann–Whitney *U* test was used to analyze differences between two independent variables for normally and non-normally distributed data, respectively. The paired *t* test was used to identify differenced in paired samples. The Bravais-Pearson correlation coefficient was used to test correlations between two variables.

The data of microbial alpha diversity showed normality and equality of variances and were, therefore, analyzed by two-way ANOVA to assess main effects and interactions between genotype (controls: Cre⁻ and Nav1.8-cre/DTA: Cre⁺) and zeitgeber time (ZT) as between-subject factors in mice fed the same diet. Multiple comparisons using Bonferroni´s *post hoc* test were performed when interactions were detected.

Regarding the DRF dataset, the evaluation of the community structure across groups was performed through principal coordinates analysis (PCoA) (phyloseq::ordinate function and "bray" distance). For differential abundance analysis of ASVs across groups, the edgeR v.1.38 R package[44] was applied using trimmed mean of M values (TMM) data normalization, quasi-likelihood F-tests and Bonferroni *post hoc* correction. Pairwise comparisons between the experimental groups were undertaken to identify the main effects and interactions, making contrasts between genotype (Cre⁻ and Cre⁺) and prandial condition (ZT12, fasting; ZT18, 6 h of refeeding, and ZT24, 12 h of refeeding) for CD- or HFHSD-fed mice.

Identification of oscillating signals in the microbiota of the *ad libitum* dataset was assessed with the web application DiscoRhythm (https://disco.camh.ca). The dataset was divided into two independent groups (Cre⁻ CD and Cre⁺ CD) and the JTK approach was selected for oscillation detection with Benjamini and Hochberg multiple testing correction.

The univariate $t$-test was applied to study, individually, the variations in bile acids according to genotype ($Cre^-$ and $Cre^+$), diet (CD or HFHSD) and prandial condition.

Sample sizes and number of replicates are provided in the figure legends for each analysis.

## Reporting summary

Further information on research design is available in the Nature Portfolio Reporting Summary linked to this article.

## Data availability

Raw data from 16 S rRNA amplicon sequencing has been deposited at European Nucleotide Archive (ENA). Accession number: PRJEB58911. The procedures for obtaining source data can be found in the general guide on ENA data retrieval. Source data for the graphs and charts are provided as Supplementary Data 1. Any remaining information can be obtained from the corresponding author upon reasonable request.

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

## Acknowledgements

We thank Inmaculada Noguera for excellent technical assistance and the animal technicians for animal care. This study received funding from the European Union Horizon 2020 research and innovation program under the Marie Sklodowska-Curie grant agreement No. 797297 (MRP), the Spanish Ministry of Science and Innovation (Grant PID2020-119536RB-I00) and the European Commission – NextGenerationEU, through the CSIC Interdisciplinary Thematic Platform (PTI +) NEURO-AGINGI+ (PTI-NEURO-AGING + )". "Severo Ochoa" grant of National Agency for Research (AEI)- Ministry of Science and Innovation (Ref. CEX2021-001189-S) is also acknowledged.

## Author contributions

Conceptualization: M.R.P. and Y.S.; Data curation: C.B.V.; Formal analysis: M.R.P., T.R., C.J.G.; Funding acquisition: M.R.P. and Y.S.; Investigation and Methodology: M.R.P., C.B.V., and I.L.A.; Supervision: M.R.P. and Y.S.; Writing - original draft: M.R.P. and C.J.G.; Writing - review & editing: Y.S. and F.A.T.B.; Project administration: M.R.P. and Y.S.

## Competing interests

The authors declare no competing interests.

## Additional information

**Supplementary information** The online version contains Supplementary Material available at https://doi.org/10.1038/s42003-024-05905-3.

