## [Peer Review File · Communications Biology]

Reviewers' comments:

Reviewer #1 (Remarks to the Author):

In this study, Bullich-Vilarrubias and colleagues assessed the requirement of Nav1.8 neurons in feeding, intestinal immunity, circadian metabolic parameters, and gut microbiota in the mouse. Briefly, they found that mice lacking Nav1.8 neurons to suffer from a wide range of immuno-metabolic anomalies including hyperphagia and intestinal immune dysregulation. Overall, this is a carefully done metabolic profiling that raises a number of interesting questions on the role of nociceptors in metabolism. Please find below this reviewer's comments:

INTRODUCTION. Overall, the introduction is clear enough. One minor comment: Line 40. In fact, Nav1.8 neurons of the nodose ganglia also respond to noxious and inflammatory stimuli but, unlike their spinal counterparts, do not convey pain.

METHODS. The authors provided detailed methods and carefully validated their mouse model. Could the authors clarify whether experimental mice always included heterozygous for Nav1.8-Cre (not homozygous)? Were mice bred in a conventional or barrier facility?

RESULTS and CONCLUSION. The result section is well written and detailed. Overall, the data appear to be consistent with the initial hypothesis and provide a very comprehensive analysis of the immuno-metabolic profile of Nav1.8 neurons-deficient animals. However, I find Figure 7 challenging to navigate personally due to an overwhelming amount of information.

GENERAL COMMENTS: 1. What could explain the lack of feeding phenotype at 10 weeks of age compared to older animals? 2. Based on the available literature, do the authors know if systemic analgesics exert immuno-metabolic effects consistent with the data reported in this study?

Reviewer #2 (Remarks to the Author):

Clara Bullich-Vilarrubias et al. has reported that deleting Nav1.8 positive cells affects gut hormones secretion, microbiota composition, and intestinal immunity, thus affecting feeding behavior and body weight. The purpose of this paper is essential, but the manuscript was written complicated to read. My concerns are as follows.

1. It seems that body weight between Cre⁻ and Cre⁺ at 5 weeks of ad-lib feeding even in CD (Fig. 6b) (not Bw gain in Fig. 6a). The authors should investigate the mechanism that creates this difference. Is this because of an increase in food intake or a decrease in energy expenditure? What happens if ad-lib feeding is continued until they become 20 weeks of age? Is the difference in food intake in Fig. 1a-b and deltaBw in Fig. 1c at 20 weeks of age due to the LRF and DRF?

2. LRF and DRF did not affect body weight, suggesting that the rhythm of microbiota composition and bile acids are not crucial for regulating food intake and body weight. The authors should prove the importance of them.

3. It was not easy to understand the protocol even though it was written in the supplementary Fig. 1. The experiment was not done step by step, so the readers would have many questions. For example, Fig. 1a shows the result of food intake at 20 weeks old after the LRF and DRF. I wondered why the authors did different types of restricted feeding and measured food intake in ad-lib. The experiment should first be done in the ad-lib and measure food intake and body weight until 20 weeks, then do only LRF, then only DRF, and then LRF+DRF. Otherwise, it's too complicated, and readers can't understand the purpose of the experiments.

We appreciate very much the helpful and constructive comments we have received from the Reviewers and their contribution to improving the report of our results, as well as the overall quality of our manuscript.

Accordingly, we have carefully addressed the issues raised by the referees and substantially modified the manuscript by clarifying our experimental approach, considering the Editor's and Reviewer's opinions.

Hereby we address point by point all aspects highlighted by the reviewers. Comments corrections are highlighted in yellow in the revised main text and revised supplemental information.

Reviewer #1 (Remarks to the Author):

In this study, Bullich-Vilarrubias and colleagues assessed the requirement of Nav1.8 neurons in feeding, intestinal immunity, circadian metabolic parameters, and gut microbiota in the mouse. Briefly, they found that mice lacking Nav1.8 neurons to suffer from a wide range of immuno-metabolic anomalies including hyperphagia and intestinal immune dysregulation. Overall, this is a carefully done metabolic profiling that raises a number of interesting questions on the role of nociceptors in metabolism. Please find below this reviewer's comments:

INTRODUCTION. Overall, the introduction is clear enough. One minor comment: Line 40. In fact, Nav1.8 neurons of the nodose ganglia also respond to noxious and inflammatory stimuli but, unlike their spinal counterparts, do not convey pain.

ANSWER: To address the reviewer's comment we have deleted the sentence "*are required to detect noxious thermal, mechanical and inflammatory stimuli*" to highlight main functions of spinal and vagal Nav1.8+ sensory neurons, independently the stimuli they respond. See modifications in line 45 of the revised main text

METHODS. The authors provided detailed methods and carefully validated their mouse model. Could the authors clarify whether experimental mice always included heterozygous for Nav1.8-Cre (not homozygous)?

ANSWER: We purchased mice expressing Cre recombinase (strain: B6.129-Scn10atm2(cre)Jnw/H) from EMMA-infrafrontier repository. This repository only had cryo-preserved heterozygous gametes. They provided to us live heterozygous animals by conducting a rederivation process.

The generation of homozygous mice carrying Cre recombinase in our animal facility, by crossing heterozygous mice, would have been extremely costly and time-consuming. Alternatively, we crossed homozygous floxed DTA female mice, obtained from Jackson Laboratory, with heterozygous male mice carrying the *Cre* gene in *Nav1.8* promoter. This strategy also deletes neurons expressing Nav1.8.

Were mice bred in a conventional or barrier facility?

ANSWER: Mice were bred in barrier facility using ventilated racks and cages. These details are included in lines 539-540 of the revised main text

RESULTS and CONCLUSION. The result section is well written and detailed. Overall, the data appear to be consistent with the initial hypothesis and provide a very comprehensive analysis of the immuno-metabolic profile of Nav1.8 neurons-deficient animals. However, I find Figure 7 challenging to navigate personally due to an overwhelming amount of information.

ANSWER: We really appreciate the suggestion of the reviewer and have simplified Figure 7 (now Figure 8) and changed the text referred to this figure in accordance (lines 353-360 and 1104-1111)

GENERAL COMMENTS:

1. What could explain the lack of feeding phenotype at 10 weeks of age compared to older animals?

ANSWER: We cannot provide a precise explanation for this, but we think that this result could be a phenotype that appears in the long term or could be a consequence of the previous time-restricted feeding regimen, as suggested by reviewer 2.

In response to reviewer 2 comments concerning the difficulties to understand the experimental design, we have reordered the figures which are now presented step by step according to the actual schedule of the experiments (see Supplementary Figure 1b). Due to the new organization, food intake comparison between ages is no longer shown. Nevertheless, we comment on the food intake discrepancy between our study and the work conducted by Udit et al 2017 in which food intake was not especially affected by the loss of Nav1.8+ neurons in the discussion of the revised version (lines 382-386)

2. Based on the available literature, do the authors know if systemic analgesics exert immuno-metabolic effects consistent with the data reported in this study?

ANSWER: Blocking Nav1.8+ neurons reduces visceral pain and referred hyperalgesia in mice

(Laird, J. M. A., Souslova, V., Wood, J. N. & Cervero, F. Deficits in visceral pain and referred hyperalgesia in Nav1.8 (SNS/PN3)-null mice. *J Neurosci* **22**, 8352–8356 (2002)). Using the same animal model (Nav1.8-cre/DTA mice), it has been demonstrated that Nav1.8-expressing neurons are required to mediate cold, mechanical, and inflammatory pain (Abrahamsen, B. et al. *The Cell and Molecular Basis of Mechanical, Cold, and Inflammatory Pain. Science* **321**, 702–705 (2008)). On the other hand, opioid-induced analgesia is partially mediated by Mu opioid receptors in Nav1.8+ (Weibel, R. et al. *Mu Opioid Receptors on Primary Afferent Nav1.8 Neurons Contribute to Opiate-Induced Analgesia: Insight from Conditional Knockout Mice. PLOS ONE* **8**, e74706 (2013)). Accordingly, as suggested the reviewer, Nav1.8-silencing strategies to induce systemic analgesia could drive similar immuno-metabolic effects reported in our study. Although it is an interesting question, we have not found studies investigating the role of Nav1.8+ neurons in analgesia/pain and their function regulating immune and energy homeostasis altogether. Nevertheless, throughout the whole discussion we mention available literature addressing the function of these neurons on the immune response (Sugisawa et al 2022, Talbot et al 2015, Lai et al 2020) and on energy homeostasis (Udit et al 2017, Mansuy-Aubert et al 2015)

Reviewer #2 (Remarks to the Author):

Clara Bullich-Vilarrubias et al. has reported that deleting Nav1.8 positive cells affects gut hormones secretion, microbiota composition, and intestinal immunity, thus affecting feeding behavior and body weight. The purpose of this paper is essential, but the manuscript was written complicated to read. My concerns are as follows.

1. It seems that body weight between Cre- and Cre+ at 5 weeks of ad-lib feeding even in CD (Fig. 6b) (not Bw gain in Fig. 6a). The authors should investigate the mechanism that creates this difference. Is this because of an increase in food intake or a decrease in energy expenditure?

ANSWER: Although we identified a main effect of the genotype on the body weight trajectory in mice fed a CD (data are now included in Supplementary Fig. 3a, left graph), controls and Nav1.8-cre/DTA mice fed CD showed similar body weight gain (Supplementary Fig. 3a, right graph) We also provide to the reviewer the Two-way ANOVA's test conducted on body weight gain after 5 weeks of *ad-libitum* feeding (see graph and table with statistics shown below). We identified an interaction between diet and genotype. *Post-hoc* Bonferroni's test indicates significant differences between the body weight gain of control mice fed CD and control mice fed HFHSD (* $p < 0.05$). Differences between controls and Nav1.8-cre/DTA mice were only identified under HFHSD (### $p < 0.001$) but not under CD ($P = 0.7094$)

ANOVA table	F (DFn, DFd)	P value
Interaction	F (1, 28) = 5,647	P=0,0246
diet	F (1, 28) = 3,133	P=0,0876
genotype	F (1, 28) = 21,68	P<0,0001
Bonferroni's test		
	Adjusted P Value	
CD:Cre- vs. CD:Cre+		0,7094
CD:Cre- vs. HFHSD:Cre-		0,0127
CD:Cre- vs. HFHSD:Cre+		0,3051
CD:Cre+ vs. HFHSD:Cre-		0,0006
CD:Cre+ vs. HFHSD:Cre+		>0,9999
HFHSD:Cre- vs. HFHSD:Cre+		0,0002

To unravel the mechanisms involved in the genotype-related body weight differences under HFHSD-feeding *ad-libitum*, we generated another cohort of mice (Nav1.8-cre/DTA and controls) that were fed HFHSD *ad libitum* for 8 weeks. Experimental design and methods are explained in lines 581-587 and lines 647-654 of the main text and graphically represented in Supplementary Fig.1a of the revised manuscript. Results are described in lines 99-109 and discussed in lines 362-372. It should be noted that we have not included the CD-fed groups due to the limited number of offspring that included mice of each genotype to conduct this experiment.

In brief, we ruled out that the reduced weight gain of Nav1.8-cre/DTA was due to lower caloric intake since, contrary to be reduced, food intake tended to be increased in mice lacking Nav1.8+ neurons (Fig. 1b). Based on previous studies (V. Mansuy-Aubert, L. Gautron, S. Lee, A. L. Bookout, C. Kusminski, K. Sun, Y. Zhang, P. E. Scherer, D. J. Mangelsdorf, J. K. Elmquist, *Elife* 2015, 4, e06667.), we investigated whether the ablation of Nav1.8+ neurons induces higher energy expenditure by measuring thermogenesis markers (UCP1) in brown adipose tissue (BAT) through immunohistochemistry. In line with Mansuy-Aubert et al 2015 findings we identified higher UCP1 expression in BAT of Nav1.8-cre/DTA although without reaching statistical significance ($p = 0.08$)

What happens if ad-lib feeding is continued until they become 20 weeks of age?

ANSWER: To respond to the reviewer's comment, we provide data from another work that is currently under revision in Molecular Nutrition & Food Research journal (ID_mnfr.202300474.R1: *The ablation of sensory neurons expressing the Nav1.8 sodium channel improves glucose homeostasis and amplifies the GLP-1 signaling in obese female mice*).

Using the same mouse model (Nav1.8-cre/DTA and control littermates), in the aforementioned study we identified that Nav1.8+ neurons exert sex-dependent effects on the metabolic phenotype. In brief, these neurons are required for controlling glucose homeostasis in females while for regulating body weight under an obesogenic diet in males. Similar to that observed in

our study, some male mice did not survive to longer periods of obesogenic diet consumption, which did not occur in females. Graphs are provided below:

Possible explanations of this survival rate in male mice fed HFHSD *ad libitum* are discussed in our other article (*ID_mnfr.202300474.R1*). Please see italicized paragraph below:

Importantly, some Nav1.8-cre/DTA male mice fed HFHSD had ulcerative dermatitis with skin lesions similar to those shown by Nav1.8-TSC2KO mice (M. Brazill, D. Shin, K. Magee, A. Majumdar, I. R. Shen, V. Cavalli, E. L. Scheller, Mol Metab 2023, 68, 101664). Although we did not perform behavioral analyses in Nav1.8-cre/DTA male mice, these skin lesions could be a consequence of an itch-related behavior as exhibited by Nav1.8-TSC2KO mice (M. Brazill, D. Shin, K. Magee, A. Majumdar, I. R. Shen, V. Cavalli, E. L. Scheller, Mol Metab 2023, 68, 101664). This sickness-related phenotype was only detected in males fed an obesogenic diet and finally caused death in our model. In agreement with other findings, these observations in Nav1.8-cre/DTA mice could reflect their inability to resolve the lesion-related inflammation due to defective immune response especially under high fat diet, as previously suggested (S. Udit, M. Burton, J. M. Rutkowski, S. Lee, A. L. Bookout, P. E. Scherer, J. K. Elmquist, L. Gautron, Mol Metab 2017, 6, 1081–1091).

In the present work, we also stated that 3 weeks of HFHSD-feeding *ad-libitum* after time-restricted feeding importantly reduced the survival of Nav1.8-cre/DTA mice (5 dead mice out 7) (lines 229-231). We also discussed the reduced survival in lines 372-379

Is the difference in food intake in Fig. 1a-b and delta Bw in Fig. 1c at 20 weeks of age due to the LRF and DRF?

As suggested by the reviewer, food intake differences between 10- and 20-weeks old mice could

be explained by the previous time-restricted feeding. In response to comment 3 (see below) we have reordered figures which are now presented step by step according to the actual schedule of the experiments. Therefore, comparisons between ages are no longer shown. In addition, in lines 382-391 we now explain the discrepancy between our study and the study conducted by Udit et al 2017 in which food intake is not especially affected by the loss of Nav1.8+ neurons.

2. LRF and DRF did not affect body weight, suggesting that the rhythm of microbiota composition and bile acids are not crucial for regulating food intake and body weight. The authors should prove the importance of them.

ANSWER: We agree with the reviewer that whether gut microbiota changes due to the absence of Nav1.8+ neurons are essential for immune and/or enteroendocrine intestinal elements and so for body weight and food intake should be proven. Indeed, this is stated in discussion (lines 529-530). Nevertheless, we would like to indicate that, under DRF we found changes associated with the loss of Nav1.8+ neurons. Indeed, under this regimen, Nav1.8-cre/DTA mice showed increased food intake (ZT12-ZT18), body weight loss and body weight gain and reduced food intake suppression after oral Ensure administration (see Figure 2 of the revised article). Variations on gut microbiota composition could be involved in all these aforementioned changes. This is suggested but not provided as conclusion

3. It was not easy to understand the protocol even though it was written in the supplementary Fig. 1. The experiment was not done step by step, so the readers would have many questions. For example, Fig. 1a shows the result of food intake at 20 weeks old after the LRF and DRF.

ANSWER: We understand the reviewer's concern, thus the revised text now presents figures step by step as the experiment was conducted.

Therefore, we first present the role of Nav1.8-expressing neurons on body weight gain in response to HFHSD-feeding *ad libitum* (Figure 1). We then show the function of Nav1.8+ neurons on the short-term control of food intake and body weight variations when mice conducted repeated fasting-refeeding cycles (Figure 2). Pre- and postprandial changes of the gut microbiota composition and bile acids due to the loss of Nav1.8+ neurons are shown in Figure 3 and Figure 4, respectively, as key elements regulating energy homeostasis. The role of Nav1.8+ neurons on daily gut microbiota rhythms are also shown under *ad libitum* regimen in Figure 5, but only in mice fed CD since Nav1.8-cre/DTA mice did not survive to chronic HFHSD consumption. In mice fed CD *ad libitum*, the function of Nav1.8+ neurons on maintaining pre- and postprandial gut hormone levels in plasma is shown Figure 6, as important elements controlling energy balance. Finally, whether the absence of Nav1.8+ neurons impact of daily

food intake of mice fed *ad libitum* CD is investigated in Figure 7. Figure 8 summarize main findings of the study.

We believe that this new order will facilitate the reader's understanding since results follow steps shown in supplementary figure 1. Changes in the text concerning new order of figures are highlighted in yellow.

I wondered why the authors did different types of restricted feeding and measured food intake in ad-lib.

ANSWER: To facilitate reader's understanding the revised text now includes a short explanation of why restricted feeding was conducted and why mice were again switched to *ad libitum* regimen. Our arguments are provided to the reviewer below:

Lines 111-119: Since the loss of Nav1.8-expressing neurons tended to dysregulate food intake and metabolic routes of fat utilization, we submitted mice to daily nutritional challenges to further explore the role of these neurons in the short-term control of food intake and body weight. Accordingly, we generated another cohort of mice (controls or Nav1.8-cre/DTA) that were first fed *ad libitum* with either control diet (CD) or HFHSD, and then switched to 12 h of fasting followed by refeeding (12 h of restricted feeding) first in the light phase (LRF) for 3 weeks and then in the dark phase (DRF) for 3 weeks to explore the influence of feeding time or light/dark cues on feeding behavior and body weight variations (**Supplementary Fig. 1b**).

Lines 170-177: The gut microbiota shows food-entrainable diurnal oscillations¹⁴ and, through its interaction with the diet, modulates feeding patterns by governing enteroendocrine signaling of the gut-brain axis¹⁵⁻¹⁷. In addition, innervations of the gastrointestinal tract establish a complex connection with the gut microbiota¹⁸. On these bases, we explored whether the ablation of Nav1.8+ neurons affects the gut microbiota composition with a particular emphasis in its pre- and postprandial oscillations during DRF, which in turn could affect feeding behavior.

Lines 224-227: We then assess whether Nav1.8-expressing neurons are also involved in the daily fluctuations of the gut microbiota when mice have free access to food. Thus, after time-restricted feeding, mice were again switched to *ad libitum* regimen during 3 weeks.

Lines 287-292: Next, we assessed whether the altered daily rhythms of the gut microbiota of Nav1.8-cre/DTA mice were aligned with altered secretion of gut hormones as primary drivers of short-term control of food intake.

At the end of the experiment, we measured gut hormones during fasting and after an oral load of Ensure® (Fig. 6a-f) in mice fed CD *ad libitum*, since, as mentioned above, most Nav1.8-cre/DTA mice did not survive a final period of *ad libitum* HFHSD feeding.

Based on the reviewer's concerns, it seems difficult to draw clear conclusions from our manuscript. Therefore, we have included a brief conclusion in each subsection of the results section in lines 160-167 lines 278-283, and lines 345-349.

We have also simplified the subtitles in the results section.

The experiment should first be done in the ad-lib and measure food intake and body weight until 20 weeks, then do only LRF, then only DRF, and then LRF+DRF. Otherwise, it's too complicated, and readers can't understand the purpose of the experiments.

ANSWER: In line with the reviewer's comment we have generated another cohort of mice to better explore the role of Nav1.8+ neurons in body weight control in response to HFHSD intake *ad libitum* (see new Figure 1 and results described in lines 99-110)

As the reviewer proposes, we could perform LRF and DRF experiments separately, but this would significantly increase the number of animals needed. Since mice were generated by crossing two strains of mice, one of them obtained after rederivation process from cryo-preserved sperm, the number of litters required to obtain sufficient number of animals to conduct a total of 4 experimental procedures would be excessive and probably not feasible in our animal facility. In addition, females are not used in the study and so a large number of animals would have to be sacrificed. We believe that the longitudinal approach shown in our study is equally valid since address similar scientific questions using lower number of mice in line with 3R of animal ethics.

We would like the reviewer consider the new experiment we have done (Fig. 1) and the reorganization of figures of the revised article as an improvement that facilitates the reader's comprehension.

REVIEWERS' COMMENTS:

Reviewer #1 (Remarks to the Author):

My comments have all been addressed with satisfaction.

Reviewer #2 (Remarks to the Author):

The paper is still challenging to read, at least for me. The experiment must be well-designed to be published in a prestigious journal like Communications Biology.

The story comes in a lot of directions. First, they showed a big change in the body weight of Nav1.8-cre/DTA mice during HFHSD, then food intake during restricted feeding of HFHSD, then bile acid and microbiome in CD, and food intake of CD. It's better to be a simple story. Readers may wonder why Nav1.8-cre/DTA mice decrease body weight during HFHSD, why the authors started LRF and DRF, and why they didn't measure gut hormones in HFHSD.

The most significant changes in the body weight are Fig1 and Fig2f (from adlib to LRF in Cre- HFHS). But the food intake can not explain the change. Digestion of the food in the gut and absorption of the nutrients from the intestines may have some trouble in Nav1.8-cre/DTA mice.

Minor comments

1. Are the groups' colors correct in Fig2a DRF? There are two Cre-CD and Cre-HFHSD in ZT12-18.
2. Fig2b, DRF ZT should be ZT12-14?
3. Fig2c, please check the order of results. Why is it different from other Figs?
4. Why did Nav1.8-cre/DTA mice of CD not change their food intake in DRF in Fig1? They increased food intake in the dark phase (Fig. 7) and altered gut hormone levels and hypothalamic alpha-Msh.

REVIEWERS' COMMENTS:

We appreciate the comments we have received from the Reviewer and we addressed the issues raised and modified the manuscript to facilitate understanding.

Corrections to the reviewer's comments are highlighted in yellow in the main revised text

Reviewer #2 (Remarks to the Author):

The paper is still challenging to read, at least for me.

In order to facilitate the understanding of the main findings, we have reorganized the content of the discussion (see corrections highlighted in yellow) in the following manner: First, we expose that the absence of Nav1.8+ neurons impacts on the control of body weight under ad libitum and dark-restricted feeding conditions. Second, we discuss aspects concerning how the ablation of Nav1.8+ neurons influences the control of food intake. Third, we state the uncoordinated regulation of feeding behaviour and body weight, which led us to explore the possible mechanisms involved independent of food intake. The mechanisms we discuss are focused on energy-dissipating metabolic pathways, supported by our results from the thermogenic markers in BAT, and by other studies from our group in Nav1.8-ablated mice (see reference 24) reporting reduced food efficiency and postprandial lipid uptake.

Then we discuss how changes in intestinal endocrine and microbial signals induced by the ablation of Nav1.8+ neurons, as relevant components of the gut-brain axis, could contribute to the observed phenotype.

The experiment must be well-designed to be published in a prestigious journal like Communications Biology. The story comes in a lot of directions. First, they showed a big change in the body weight of Nav1.8-cre/DTA mice during HFHSD, then food intake during restricted feeding of HFHSD, then bile acid and microbiome in CD, and food intake of CD. It's better to be a simple story.

We believe that we have explained the experiments in chronological order.

To facilitate the understanding, explanations of how each result leads to the next scientific question are provided throughout the "results" section. Explanations of why certain analyses were conducted in CD-fed mice are also included.

Here we provide to the reviewer line numbers where one can find all these explanations and a brief description of why we perform the next analysis:

-100-111: Under an obesogenic diet we found resistance to weight gain in Nav1.8-cre/DTA mice that could not be explained by food intake results, and so, we explored thermogenesis in BAT as alternative mechanism involved.

-112-115: Based on the uncoordinated regulation of food intake and body weight, 3 weeks of 12h of feeding restriction was conducted as nutritional challenge, first during the light phase and second during the dark phase, to explore food intake and body weight variations. This is a procedure commonly used to challenge energy homeostasis that allows exacerbating hunger signals and catabolic pathways helping to uncover effects that could not appear under ad libitum conditions.

-127-130: Here we explain the parameters we measured under the food restriction regimen and why we measured food intake every 6h.

-142-144: Since Nav1.8+ neurons ablated mice had exacerbated hunger, we explored whether these mice have impaired satiety by evaluating food intake suppression in response to an oral load of Ensure®

-148: this is a line concerning the exploration of body weight variations under the food restriction regimen

161-168: Based on the experimental design explained and justified in the aforementioned lines, here we provide a conclusion of main results

-172-178: In these lines we justify why gut microbiota (composition and metabolites) was explored. Since the molecular events occurring before and after a meal are especially relevant in the study of food intake control, in line with hunger and satiety sensation, we explored pre and postprandial variations of the gut microbiota. These variations can be more easily controlled under a food restriction regime since synchronizes the mice to start eating

-225-233: Since we also considered relevant to analyse daily gut microbiota variations under a regimen of ad libitum feeding, animals were switched to ad libitum conditions. In these lines, we also state why analyses were conducted only in CD-fed mice. Indeed, under HFHSD, the switch from restricted to *ad libitum* feeding importantly reduced the survival of mice lacking Nav1.8+ neurons.

-288-293: Here we explain why gut hormones were explored and why these hormones were only analysed in CD-fed mice.

For all of this, we considered that this work is properly designed since all the procedures conducted respond to a scientific question.

Readers may wonder why Nav1.8-cre/DTA mice decrease body weight during HFHSD, why the authors started LRF and DRF, and why they didn't measure gut hormones in HFHSD.

The explanation for reduced body weight during HFHSD is provided in the results section concerning the thermogenesis-related protein UCP1 (lines 107-111) and in the discussion section where, besides mentioning our results regarding UCP1 (lines 391-393), we cite other supporting results from another study from our group in Nav1.8-ablated mice (see reference 24) (lines 400-403), in which we reported reduced food efficiency and lipid uptake.

For the explanation for the LRF and DRF, see lines 112-120 in the main text. For further details, please see our reply to the previous comment.

The explanation of why we did not measure gut hormones on HFHSD feeding is provided in lines 291-293 of the main text.

The most significant changes in the body weight are Fig1 and Fig2f (from adlib to LRF in Cre- HFHS). But the food intake cannot explain the change. Digestion of the food in the gut and absorption of the nutrients from the intestines may have some trouble in Nav1.8-cre/DTA mice.

We agree with the reviewer that food intake cannot explain body weight gain differences associated to the genotype. In line with the reviewer concern, we now specifically indicate in the discussion that the absence of Nav1.8+ neurons disrupts the coordinated control of body weight gain and feeding behaviour (see lines 385-388). For clarity, we have modified the text (discussion section) to suggest that resistance to HFHSD-induced weight gain could be explained by an overstimulation of energy-dissipating metabolic pathways as supported by UCP1 analysis in BAT (see parts highlighted in yellow between lines 388-400).

In addition, we have included a recent publication from our group in line with the reviewer's comment concerning the impaired intestinal nutrient absorption in Nav1.8-cre/DTA (see lines 400-403).

Minor comments

1. **Are the groups' colors correct in Fig2a DRF? There are two Cre-CD and Cre-HFHSD in ZT12-18.**

2. **Fig2b, DRF ZT should be ZT12-14?**

3. **Fig2c, please check the order of results. Why is it different from other Figs?**

Concerning comments related to Figure 2, we appreciate the reviewer's observations. All the mistakes have been corrected

-Indeed, in Fig 2a, data from groups Cre+ ZT12-ZT18 (CD and HFHSD) and Cre-ZT18-ZT24 (CD and HFHSD) were incorrectly located. This is now corrected in revised Fig 2a

-In Fig 2b DRF we have changed ZT0-ZT2 by ZT12-ZT14

-We have also changed the order of the results in Fig 2c in line with the rest of the graphs

4. **Why did Nav1.8-cre/DTA mice of CD not change their food intake in DRF in Fig1? They increased food intake in the dark phase (Fig. 7) and altered gut hormone levels and hypothalamic alpha-Msh.**

Fig 1 shows data Nav1.8-cre/DTA mice fed HFHSD and not fed CD.

We provide a response on the assumption that the reviewer refers to comparisons between data from Fig 2a with data from Fig 7a.

The potential discrepancies stated by the reviewer could be explained by the different type of food intake regimen conducted in each figure. Thus, food intake could be differently regulated under a 12h-food restricted regimen (Fig. 2) and under ad libitum food intake (Fig. 7)

Still, since, Fig. 2 does not include 12h-food intake of mice (i.e. ZT12-ZT24), to address the reviewer's concern and to demonstrate that our data do not show discrepancies, 12h-food intake data from CD-fed groups in Fig.2a (i.e. food intake from ZT12 to ZT24) were analysed with Student's t test as in Fig. 7a (see figure below).

Supporting the coherence of our data, the analysis restricted to CD-fed groups under DRF regimen demonstrated that Nav1.8-Cre/DTA ate more than their control littermates as also shown in Fig 7a in an ad libitum regimen. This result is shown in the figure below, which is not included in Fig. 2 since it aims to highlight that the genotype effect in 6h-food intake (ZT12-ZT18, ZT18-ZT24) was more relevant under HFHSD than in CD (two-way ANOVA followed by Bonferroni's post hoc test).

$t = 2.29, df = 14, p = 0.03$